# Flow Priors for Linear Inverse Problems via Iterative Corrupted Trajectory Matching

**Yasi Zhang**
UCLA
yasminzhang@ucla.edu

**Peiyu Yu**
UCLA
yupeiyu98@g.ucla.edu

**Yaxuan Zhu**
UCLA
yaxuanzhu@g.ucla.edu

**Yingshan Chang**
CMU
yingshac@andrew.cmu.edu

**Feng Gao**[*]
Amazon
fenggo@amazon.com

**Ying Nian Wu**
UCLA
ywu@stat.ucla.edu

**Oscar Leong**
UCLA
oleong@stat.ucla.edu

## Abstract

Generative models based on flow matching have attracted significant attention for their simplicity and superior performance in high-resolution image synthesis. By leveraging the instantaneous change-of-variables formula, one can directly compute image likelihoods from a learned flow, making them enticing candidates as priors for downstream tasks such as inverse problems. In particular, a natural approach would be to incorporate such image probabilities in a maximum-a-posteriori (MAP) estimation problem. A major obstacle, however, lies in the slow computation of the log-likelihood, as it requires backpropagating through an ODE solver, which can be prohibitively slow for high-dimensional problems. In this work, we propose an iterative algorithm to approximate the MAP estimator efficiently to solve a variety of linear inverse problems. Our algorithm is mathematically justified by the observation that the MAP objective can be approximated by a sum of $N$ "local MAP" objectives, where $N$ is the number of function evaluations. By leveraging Tweedie's formula, we show that we can perform gradient steps to sequentially optimize these objectives. We validate our approach for various linear inverse problems, such as super-resolution, deblurring, inpainting, and compressed sensing, and demonstrate that we can outperform other methods based on flow matching. Code is available at https://github.com/YasminZhang/ICTM.

## 1 Introduction

Linear inverse problems are ubiquitous across many imaging domains, pervading areas such as astronomy [41, 23], medical imaging [38, 49], and seismology [35, 39]. In these problems the goal is to reconstruct an unknown image $x_* \in \mathbb{R}^n$ from observed measurements $y \in \mathbb{R}^m$ of the form:

$$y = \mathcal{A}(x_*) + \text{noise}, \tag{1}$$

where $\mathcal{A} : \mathbb{R}^n \to \mathbb{R}^m$ with $m \leq n$ is a linear operator that degrades the clean image $x_*$, and the additive noise is drawn from a known distribution. In this work, we assume the noise follows $\mathcal{N}(0, \sigma_y^2 I)$. Due to the under-constrained nature of such problems, they are typically ill-posed, i.e.,

---

[*]This work is not related to the author's position at Amazon.

38th Conference on Neural Information Processing Systems (NeurIPS 2024).

there are an infinite number of undesirable images that fit to the observed measurements. Hence, one requires further structural information about the underlying images, which constitutes our prior.

With the advent of large generative models [27, 17, 48, 40, 8, 59, 58], there has been a surge of interest in exploiting generative models as priors to solve inverse problems. Given a pretrained generator to sample from a distribution or grant access to image probabilities, one can solve a variety of inverse problems in a task- or forward model-agnostic fashion, without the need for large-scale supervision [36]. This has been successfully done for a variety of models, including implicit generators such as Generative Adversarial Networks (GANs) and Variational Autoencoders (VAEs) [4, 34], invertible generators such as Normalizing Flows [1, 54], and more recently Diffusion models [10, 43, 60].

A recent paradigm in generative modeling [48, 55, 25, 57], based on the concept of flow matching [29, 28], has made significant strides in scaling ODE-based generators to high-resolution images. Flow matching models map a simple base distribution, such as a Gaussian, to a complex, high-dimensional data distribution by defining a flow field that represents the transformation between these distributions. These generative models have demonstrated scalability to high dimensions, forming the backbone of several state-of-the-art generative models [30, 13, 56]. Moreover, flow matching models follow straighter and more direct probability paths compared to diffusion models, allowing for more efficient and faster sampling [28, 29, 13]. Additionally, due to their invertibility, flow matching models provide direct access to image likelihoods through the instantaneous change-of-variables formula [9, 18]. Given these advantages and the relatively recent application of these models to inverse problems [37, 2], we investigate their use as image priors in this work.

Leveraging knowledge about the corruption process $p(y|x)$ and a natural image prior $p(x)$, the Bayesian approach suggests analyzing the image reconstruction posterior $p(x|y) \propto p(y|x)p(x)$ to solve the inverse problem . A proven and effective method based on this approach is maximum-a-posteriori (MAP) estimation [6, 19], which maximizes the posterior to identify the image most likely to match the observed measurements:

$$\underset{x \in \mathbb{R}^n}{\operatorname{argmin}} - \log p(x|y) = \underset{x \in \mathbb{R}^n}{\operatorname{argmin}} - \log p(y|x) - \log p(x). \tag{2}$$

MAP estimation provides a single, most probable point estimate of the posterior distribution, making it simple and interpretable. This deterministic approach ensures consistency and reproducibility, which are essential in applications requiring reliable outcomes, particularly in compressed sensing tasks such as Computed Tomography (CT) [7] and Magnetic Resonance Imaging (MRI) [52]. While posterior sampling methods can offer diverse reconstructions to quantify uncertainty, they can be prohibitively slow in high-dimensions [5]. Hence, in this work, we propose to integrate flow priors to solve linear inverse problems by MAP estimation.

A significant challenge in employing flow priors for MAP estimation lies in the slow computation of the image probabilities, as it requires backpropagating through an ODE solver [47, 16, 15]. In this work, we show how one can address this challenge via Iterative Corrupted Trajectory Matching (ICTM), a novel algorithm to approximate the MAP solution in a computaionally efficient manner. In particular, we show how one can approximately find an MAP solution by sequentially optimizing a novel simpler, auxillary objective that approximates the true MAP objective in the limit of infinite function evaluations. For finite evaluations, we demonstrate that this approximation is sufficient to optimize by showcasing strong empirical performance for flow priors across a variety of linear inverse problems. We summarize our **contributions** as follows:

1. We propose ICTM, an algorithm to approximate the MAP solution to a variety of linear inverse problems using a flow prior. This algorithm optimizes an auxillary objective that partitions the flow model's trajectory into $N$ "local MAP" objectives, where $N$ is the number of function evaluations (NFEs). By leveraging Tweedie's formula, we show that we can perform gradient steps to sequentially optimize these objectives.

2. Theoretically, we demonstrate that the auxillary objective converges to the true MAP objective as the NFEs goes to infinity. We validate the correctness of our algorithm in finding the MAP solution on a denoising problem.

3. We demonstrate the utility of ICTM on a wide variety of linear inverse problems on both natural and scientific image datasets, with problems including denoising, inpainting, super-resolution, deblurring, and compressed sensing. Extensive results show that ICTM is both computationaly efficient and obtains high-quality reconstructions, outperforming other reconstruction algorithms based on flow priors.

## 2 Background

**Notation** We follow the convention for flow-based models, where Gaussian noise is sampled at timestep 0, and the clean image corresponds to timestep 1. Note that this is the opposite of diffusion models. For $t \in [0, 1]$, we denote $x_t(x_0)$ as the point at time $t$ whose initial condition is $x_0$. In this work, we use $x$ and $x_1$ interchangeably, i.e., $x_1(x_0) = x(x_0)$.

### 2.1 Flow-Based Models

We consider generative models that map samples $x_0$ from a noise distribution $p(x_0)$, e.g., Gaussian, to samples $x_1$ of a data distribution $p(x_1)$ using an ordinary differential equation (ODE):

$$dx_t = v_\theta(x_t, t) \, dt, \tag{3}$$

where the velocity field $v$ is a $\theta$-parameterized neural network, e.g., using a UNet [28, 29, 42] or Transformer [13, 51] architecture. Generative models based on flow matching [28, 29] can be seen as a simulation-free approach to learning the velocity field. This approach involves pre-determining paths that the ODE should follow by specifying the interpolation curve $x_t$, rather than relying on the MLE algorithm to implicitly discover them [9]. To construct such a path, which is not necessarily Markovian, one can define a **differentiable** nonlinear interpolation between $x_0$ and $x_1$:

$$x_t = \alpha_t x_1 + \beta_t x_0, \quad x_0 \sim \mathcal{N}(0, I), \tag{4}$$

where both $\alpha_t$ and $\beta_t$ are differentiable functions with respect to $t$ satisfying $\alpha_0 = 0$, $\beta_0 = 1$, and $\alpha_1 = 1$, $\beta_1 = 0$. This ensures that $x_t$ is transported from a standard Gaussian distribution to the natural image manifold from time 0 to time 1. In contrast, the diffusion process [48, 45, 20] induces a non-differentiable trajectory due to the diffusion term in the SDE formulation.

The idea behind flow matching is to utilize the power of deep neural networks to efficiently predict the velocity field at each timestep. To achieve this, we can train the neural network by minimizing an $L_2$ loss between the sampled velocity and the one predicted by the neural network:

$$\mathcal{L}(\theta) = \mathbb{E}_{t, p(x_1), p(x_0)} \|v_\theta(x_t, t) - (\dot{\alpha}_t x_1 + \dot{\beta}_t x_0)\|^2. \tag{5}$$

We denote the optimal (not necessarily unique) solution to $\arg\min_\theta \mathcal{L}(\theta)$ as $\hat{\theta}$. The optimal velocity field $v_{\hat{\theta}}$ can be derived in closed form and is the expected velocity at state $x_t$:

$$v_{\hat{\theta}}(x_t, t) = \mathbb{E}_{p(x_1), p(x_0)}[\dot{\alpha}_t x_1 + \dot{\beta}_t x_0 \mid x_t]. \tag{6}$$

For convenience, in the following text, we use $v_\theta$ to refer to the optimal $v_{\hat{\theta}}$. In the rest of the paper, we assume that the flow $v_\theta$ and its parameters are pretrained on a dataset of interest and fixed. We are then interested in leveraging its utility as a prior to solve inverse problems.

### 2.2 Probability Computation for Flow Priors

Denote the probability of $x_t$ in Eq. (3) as $p(x_t)$ dependent on time. Assuming that $v_\theta$ is uniformly Lipschitz continuous in $x_t$ and continuous in $t$, the change in log probability also follows a differential equation [9, 18]:

$$\frac{\partial \log p(x_t)}{\partial t} = -\mathrm{tr}\left(\frac{\partial}{\partial x} v_\theta(x_t, t)\right). \tag{7}$$

One can additionally obtain the likelihood of the trajectory via integrating Eq. (7) across time

$$\log p(x_t) = \log p(x_\tau) - \int_\tau^t \mathrm{tr}\left(\frac{\partial}{\partial x} v_\theta(x_s, s)\right) ds, \ 0 \le \tau < t \le 1. \tag{8}$$

## 3 Method

In this work, we aim to solve the MAP estimation problem in Eq. (2) where $p(x)$ is given by a pretrained flow prior. We first discuss in Section 3.1 how the MAP problem could, in principle, be solved via a latent-space optimization problem. As we will see, this problem is challenging to solve

computationally due to the need to backpropagate through an ODE solver. To overcome this, we show in Section 3.2 that the ideal MAP problem can be approximated by a weighted sum of "local MAP" optimization problems, which operates by partitioning the flow's trajectory to a reconstructed solution. We then introduce our ICTM algorithm to sequentially optimize this auxiliary objective. Finally, in Section 3.3, we experimentally validate that our algorithm finds a solution that is faithful to the MAP estimate in a simplified setting where the globally optimal MAP solution is known.

### 3.1 Flow-Based MAP

Given a pretrained flow prior, one can compute the log-likelihood of $x$ generated from an initial noise sample $x_0$ via Eq. (8). Hence, to find the MAP estimate, one could equivalently optimize the initial point of the trajectory $x_0$ and return $x_1(x_0)$ where $x_0$ is found by solving

$$\min_{x_0 \in \mathbb{R}^n} \underbrace{\frac{1}{2\sigma_y^2}\|y - \mathcal{A}(x_1(x_0))\|^2}_{\text{data likelihood}} + \underbrace{\frac{1}{2}\|x_0\|^2 + \int_0^1 \operatorname{tr}\left(\frac{\partial}{\partial x}v_\theta(x_t, t)\right) dt}_{\text{prior}}, \tag{9}$$

where $x_t := x_t(x_0)$ denotes the intermediate state $x_t$ generated from $x_0$. Intuitively, this loss encourages finding an initial point $x_0$ such that the reconstruction $x_1 := x_1(x_0)$ fits the observed measurements, but is also likely to be generated by the flow.

In practice, $x_1$ and the prior term can be approximated by an ODE solver. The trajectory of $x_t = x_0 + \int_0^t v_\theta(x_t, t)dt$ can be approximated by an ODE sampler, i.e. ODESolve($x_0, 0, t, v_\theta$), where $x_0$ is the initial point, and the second and third arguments represent the starting time and the ending time, respectively. For example, with an Euler sampler, we iterate over $x_{t+\Delta t} = x_t + v_\theta(x_t, t)\Delta t$ where $\Delta t = 1/N$ and $N$ is the predetermined NFEs. After acquiring the optimal $\hat{x}_0$ by optimizing the Eq. (9), we obtain the MAP solution $x_1$ by using ODESolve($\hat{x}_0, 0, 1, v_\theta$) again.

### 3.2 Flow-Based MAP Approximation

The global flow-based MAP objective Eq. (9) is tractable for low-dimensional problems. The challenge for high-dimensional problems, however, is that optimizing Eq. (9) is simulation-based, and thus each update iteration requires full forward and backward propagation through an ODE solver, resulting in issues regarding memory inefficiency and time, making it hard to optimize [9, 15, 16, 47].

As a way to address this, we prove a result in Theorem 1 that shows that the MAP objective can be approximated by a weighted sum of $N$ local posterior objectives. These objectives are "local" in the sense that they mainly depend on likelihoods and probabilities of intermediate trajectories $x_t$ and $x_t + v_\theta(x_t, t)\Delta t$ for $t = 0, \Delta t, \dots, N\Delta t$ where $\Delta t := 1/N$. Given an initial noise input $x_0$, each local posterior objective depends on a non-Markovian **auxiliary path** $y_t = \alpha_t y + \beta_t \mathcal{A}(x_0)$ by connecting the points between $y$ and $\mathcal{A}x_0$. We prove this result for straight paths $\alpha_t = t$ and $\beta_t = 1 - t$ for simplicity, but other interpolation paths can be used. The proof is in Section A.2.

**Theorem 1.** *For $N \geq 1$, set $\gamma_i := (\frac{1}{2})^{N-i+1}$ and $\Delta t = 1/N$. Suppose $y = \mathcal{A}(x_*) + \epsilon$ where $x_* = x_1(x_0)$ with $x_0$ being the solution to Eq. (9), $\epsilon \sim \mathcal{N}(0, \sigma_y^2 I)$, and $x_t$ exactly follows the straight path $x_t = tx + (1-t)x_0$ for any timestep $t \in [0, 1]$. Suppose the velocity field $v_\theta : \mathbb{R}^n \times \mathbb{R} \to \mathbb{R}^n$ satisfies $\sup_{z \in \mathbb{R}^n, s \in [0,1]} |\operatorname{tr}\frac{\partial}{\partial x}v_\theta(z, s)| \leq C_1$ for some universal constant $C_1$. Then, there exists a constant $c(N)$[2] that does not depend on $x_0$ such that*

$$\lim_{N \to \infty} \left| \log p(x(x_0)|y) - \sum_{i=1}^N \gamma_i \hat{\mathcal{J}}_i - c(N) \right| = 0,$$

*where $\hat{\mathcal{J}}_i = \log p(x_{(i-1)\Delta t}) - \operatorname{tr}\left(\frac{\partial v_\theta(x_{(i-1)\Delta t}, (i-1)\Delta t)}{\partial x}\right)\Delta t + \log p(y_{i\Delta t}|x_{i\Delta t}).$*

This result shows that the true MAP objective evaluated at the optimal solution can be approximated by a weighted sum of objectives that depend locally at a time $t$ for the trajectory $\{x_t : t \in [0, 1]\}$. The intuition regarding $\hat{\mathcal{J}}_i$ arises from the fact that $\hat{\mathcal{J}}_i \approx \mathcal{J}_i$, where $\mathcal{J}_i$ is the local posterior distribution

$$\mathcal{J}_i = \log p(y_{i\Delta t}|x_{i\Delta t}(x_{(i-1)\Delta t})) + \log p(x_{i\Delta t}).$$

---

[2]This is given by $c(N) := \sum_{i=1}^N \gamma_i c_i - \log p(y)$. Please see the proof of Theorem 1 in Appendix A.2.

Optimizing each of these local posterior distributions in a sequential fashion captures the fact that we would like each intermediate point in our trajectory $x_{i\Delta t}$ to be likely and fit to our measurements, ideally resulting in a final reconstruction $x_1$ that satisfies this as well. The benefit of $\hat{\mathcal{J}}_i$, as we will show in the sequel, is that it is efficient to optimize.

**Discussion of assumptions:** We assume that the trajectory $\{x_t\}_t$ exactly follows the predefined interpolation path $\{\alpha_t x + \beta_t x_0\}_t$. In Section B of the appendix, we analyze this assumption and show that we can bound the deviation from the predefined interpolation path to the learned path via a path compliance measure. Moreover, we impose a regularity assumption on the velocity field $v_\theta$, effectively requiring a uniform bound on the spectrum of the Jacobian of $v_\theta$. This can be easily satisfied with neural networks using Lipschitz continuous and differentiable activation functions.

As we see in Theorem 1, one can approximate the true MAP objective via a sum of local objectives of the form

$$\hat{\mathcal{J}}_i := \underbrace{\log p(y_{i\Delta t}|x_{i\Delta t})}_{\text{local data likelihood}} + \underbrace{\log p(x_{(i-1)\Delta t}) - \text{tr}\left(\frac{\partial v_\theta(x_{(i-1)\Delta t}, (i-1)\Delta t)}{\partial x}\right)\Delta t}_{\text{local prior}}. \tag{10}$$

At first glance, $\hat{\mathcal{J}}_i$ still appears challenging to optimize, but there are additional insights we can exploit for computation. We discuss each term in $\hat{\mathcal{J}}_i$ below.

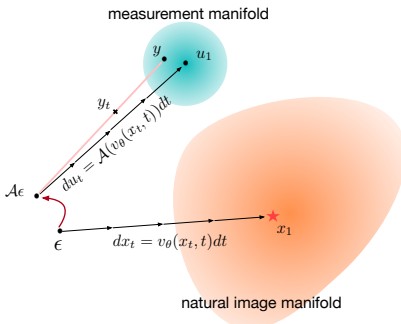

Figure 1: **Illustration of the idea of ICTM.** The corrupted trajectory $u_t := \mathcal{A}(x_t)$ follows the corrupted flow ODE $du_t = \mathcal{A}(v_\theta(x_t, t))dt$.

**Local data likelihood:** The intuition behind ICTM is that we aim to match a **corrupted** trajectory $\{u_t\}_t$ with an auxiliary path $\{y_t\}_t$ specified by an interpolation between our measurements $y$ and $\mathcal{A}(x_0)$ for each timestep $t$, defined by $y_t := \alpha_t y + \beta_t \mathcal{A}(x_0)$. The corrupted trajectory $u_t := \mathcal{A}(x_t)$ follows the **corrupted flow ODE** $du_t = \mathcal{A}(v_\theta(x_t, t))dt$. To optimize the above "local MAP" objectives, we must understand the distribution of $p(y_t|x_t)$. Generally speaking, this distribution is intractable. However, by assuming exact **compliance** of the trajectory generated by flow to the predefined interpolation path (as done in Theorem 1), we can show that $y_t|x_t \sim \mathcal{N}(u_t, \alpha_t^2 \sigma_y^2)$. This is proven in Lemma 3 in the appendix. While exact compliance of the trajectory may not hold for learned flow matching models, we show empirically that making this assumption leads to strong performance in practice. We further analyze this notion of compliance in Section B of the appendix.

**Local prior:** The approximation in Eq. (10) addresses one of the main concerns of MAP in that the intensive integral computation is circumvented with a simpler Riemannian sum. This approximation holds for small time increments $\Delta t$: $\int_t^{t+\Delta t} \text{tr}\left(\frac{\partial}{\partial x} v_\theta(x_s, s)\right) ds \approx \text{tr}\left(\frac{\partial}{\partial x} v_\theta(x_t, t)\right)\Delta t$. Note that one can additionally improve the efficiency of this term by employing a Hutchinson-Skilling estimate [44, 21] for the trace of the Jacobian matrix. However, at first glance, it appears we have simply shifted the problem to the computation of the prior at timestep $(i-1)\Delta t$. Fortunately, it is possible to derive a formula for the gradient of $\log p(x_t)$ for all timesteps $t \in [0, 1]$ using Tweedie's formula [12]. This allows us to optimize each objective $\hat{\mathcal{J}}_i$ using gradient-based optimizers. The following result gives a precise characterization of $\nabla_{x_t} \log p(x_t)$, proven in Section A.1.

**Proposition 1.** *Let $\lambda_t = \alpha_t/\beta_t$ denote the signal-to-noise ratio. The relationship between the score function $\nabla_{x_t} \log p(x_t)$ and the velocity field $v_\theta(x_t, t)$ is given by:*

$$\nabla_{x_t} \log p(x_t) = \frac{1}{\beta_t^2}\left[\left(\frac{d\log\lambda_t}{dt}\right)^{-1}\left(v_\theta(x_t, t) - \frac{d\log\beta_t}{dt}x_t\right) - x_t\right]. \tag{11}$$

In summary, we have derived an efficient approximation to the MAP objective. For our algorithm, we iteratively optimize each term $\hat{\mathcal{J}}_t$ sequentially for each $t = 0, \Delta t, \ldots, N\Delta t$, fitting our current iterate $x_t$ to induce an increment $x_{t+\Delta t}$ such that $\mathcal{A}(x_{t+\Delta t})$ fits to our auxiliary corrupted path $y_{t+\Delta t}$ while

---
**Algorithm 1** Iterative Corrupted Trajectory Matching (ICTM) with Euler Sampler
---
**Input:** measurement $y$, matrix $\mathcal{A}$, pretrained flow-based model $\theta$, NFEs $N$, interpolation coefficients $\{\alpha_t\}_t$ and $\{\beta_t\}_t$, step size $\eta$, guidance weight $\lambda$, and iteration number $K$
**Output:** recovered clean image $x_1$
  1: **Initialize** $\epsilon \sim \mathcal{N}(0, I)$, $x_0 \leftarrow \epsilon$, $t \leftarrow 0$, $\Delta t \leftarrow 1/N$
  2: **Generate** an auxiliary path $y_s = \alpha_s y + \beta_s(\mathcal{A}x_0)$ for $s \in (0, 1)$
  3: **while** $t < 1$ **do**
  4:     $x_{t+\Delta t} \leftarrow x_t + v_\theta(x_t, t)\Delta t$
  5:     **if** $t = 0$ **then**
  6:         **for** $k = 1, \cdots K$ **do**
  7:             $x_t \leftarrow x_t - \eta \nabla_{x_t} \left[ \lambda \|\mathcal{A}(x_{t+\Delta t}(x_t)) - y_{t+\Delta t}\|^2 + \frac{1}{2}\|x_t\|^2 + \mathrm{tr}\left( \frac{\partial v_\theta(x_t,t)}{\partial x} \right) \Delta t \right]$
  8:         **end for**
  9:     **else**
10:         **for** $k = 1, \cdots K$ **do**
11:             # use Eq. (11) to obtain the gradient of $\log p(x_t)$
12:             $x_t \leftarrow x_t - \eta \nabla_{x_t} \left[ \lambda \|\mathcal{A}(x_{t+\Delta t}(x_t)) - y_{t+\Delta t}\|^2 - \log p(x_t) + \mathrm{tr}\left( \frac{\partial v_\theta(x_t,t)}{\partial x} \right) \Delta t \right]$
13:         **end for**
14:     **end if**
15:     $x_{t+\Delta t} \leftarrow x_t + v_\theta(x_t, t)\Delta t$
16:     $t \leftarrow t + \Delta t$
17: **end while**
18: **return** $x_1$
---

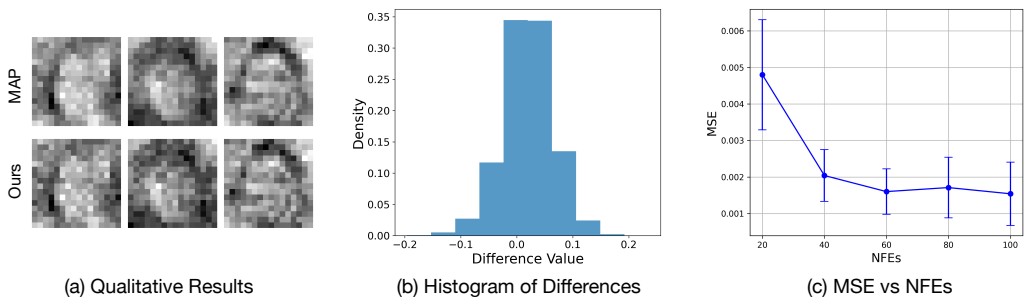

    (a) Qualitative Results         (b) Histogram of Differences         (c) MSE vs NFEs

Figure 2: Results of a toy example modeling 1,000 FFHQ faces as a Gaussian distribution. Subfigure (a) shows the qualitative results of our method; Subfigure (b) presents the histogram of the differences between ours and the true MAP; Subfigure (c) displays the MSE values as the NFEs varies.

being likely under our local prior. We call this approach Iterative Corrupted Trajectory Matching (ICTM). Our algorithm is summarized in Algo. 1. In lines 7 and 12, instead of directly optimizing the local data likelihood, we choose $\lambda$ as a new hyper-parameter to tune. We find a constant $\lambda$ works well in practice.

### 3.3 Toy Example Validation

We experimentally validate that the reconstruction found via ICTM is close to the optimal MAP solution in a simplified denoising problem where the MAP solution can be obtained in closed-form. Specifically, we fit a Gaussian distribution $\mathcal{N}(\mu, \Sigma)$ using 1,000 samples from the FFHQ dataset. Consider a denoising problem $y = x + \epsilon$ where $x \sim \mathcal{N}(\mu, \Sigma)$ and $\epsilon \sim \mathcal{N}(0, \sigma_y^2 I)$. In this case, the analytical solution to the MAP estimation problem (Eq. (2)) is $x_* = (\Sigma^{-1} + \sigma_y^{-2}I)^{-1}(\Sigma^{-1}\mu + \sigma_y^{-2}y)$. We set $\sigma_y = 0.1$. Then, we train a flow-based model on 10,000 samples from the true Gaussian distribution and showcase the deviation of our reconstruction found via ICTM to the closed-form MAP solution $x_*$ in Fig. 2. We see that ICTM can obtain a faithful estimate of the MAP solution across many samples.

Table 1: Quantitative comparison results in terms of PSNR and SSIM on the CelebA-HQ dataset. Our algorithm surpasses all other baselines across all tasks. The best values are highlighted in blue and the second-best are underlined.

| Method | Super-Resolution | | Inpainting(random) | | Gaussian Deblurring | | Inpainting(box) | |
|---|---|---|---|---|---|---|---|---|
| | PSNR | SSIM | PSNR | SSIM | PSNR | SSIM | PSNR | SSIM |
| OT-ODE | 27.46 | 0.775 | 28.57 | 0.838 | 26.28 | 0.727 | 19.80 | 0.795 |
| DPS-ODE | 27.85 | 0.791 | 29.57 | 0.872 | 25.97 | 0.704 | 23.59 | 0.758 |
| RED-Diff | 27.20 | 0.760 | 25.13 | 0.711 | 27.23 | 0.765 | 17.50 | 0.651 |
| ΠGDM | 28.33 | 0.803 | 29.98 | 0.858 | 24.30 | 0.583 | 24.10 | 0.853 |
| Ours (w/o prior) | 26.06 | 0.724 | 29.01 | 0.835 | 25.13 | 0.676 | 22.42 | 0.803 |
| Ours | 27.91 | 0.805 | 30.65 | 0.894 | 26.54 | 0.760 | 24.34 | 0.866 |

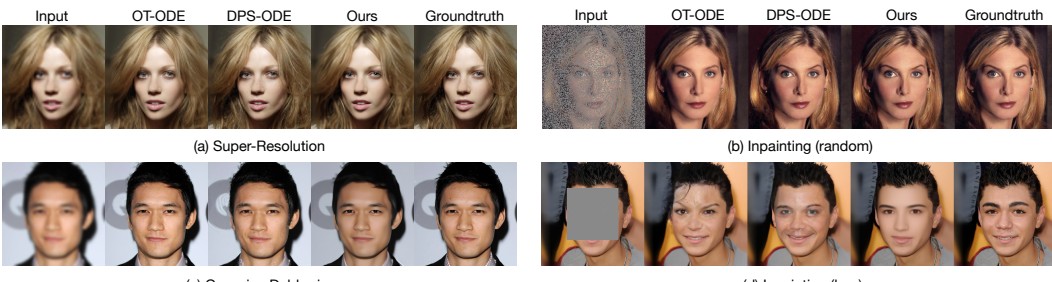

(a) Super-Resolution  (b) Inpainting (random)

(c) Gaussian Deblurring  (d) Inpainting (box)

Figure 3: Qualitative comparison results on the CelebA-HQ dataset. The reconstructions generated by our method align more faithfully with the ground truth and exhibit a higher degree of refinement.

## 4 Experiments

In our experimental setting, we use optimal transport interpolation coefficients, i.e. $\alpha_t = t$ and $\beta_t = 1 - t$. We test our algorithm on both natural and medical imaging datasets. For natural images, we utilize the pretrained checkpoint from the official Rectified Flow repository[3] and evaluate our approach on the CelebA-HQ dataset [31, 24]. We address four common linear inverse problems: super-resolution, inpainting with a random mask, Gaussian deblurring, and inpainting with a box mask. For the medical application, we train a flow-based model from scratch on the Human Connectome Project (HCP) dataset [50] and test our algorithm specifically for compressed sensing at different compression rates. Our algorithm focuses on the reconstruction faithfulness of generated images, therefore employing PSNR and SSIM [53] as evaluation metrics.

**Baselines** We compare our method with five baselines. 1) OT-ODE [37]. To our knowledge, this is the only baseline that applies flow-based models to inverse problems. They incorporate a prior gradient correction at each sampling step based on conditional Optimal Transport (OT) paths. For a fair comparison, we follow their implementation of Algorithm 1, providing detailed ablations on initialization time $t'$ in Appendix E.3. 2) DPS-ODE. Inspired by DPS [10], we replace the velocity field with a conditional one, i.e., $v(x_t|y) = v(x_t) + \zeta_t \nabla_{x_t} \log p(y|\hat{x}_1(x_t))$, where $\zeta_t$ is a hyperparameter to tune. Following the hyperparameter instruction in DPS, we provide detailed ablations on $\zeta_t$ in Appendix E.3. 3) Ours without local prior. To examine the local prior term's effectiveness in our optimization algorithm, we drop the local prior term as defined in Eq. (10) in our algorithm. In the experiments with natural images, in addition to the flow-based baselines, we have included two representative diffusion-based baselines: 4) RED-Diff [33], a variational Bayes-based method; and 5) ΠGDM [46], an advanced MCMC-based method. We also note one concurrent work, D-Flow [2], which formulates the MAP as a constrained optimization problem in their Eq. 9. As documented in their Sec. 3.4, it takes 5-10 minutes to recover each image. This is because each of its optimization step requires backpropagation through an ODE solver to compute the full log-likelihood.

[3] https://github.com/gnobitab/RectifiedFlow

Table 2: Results of compressed sensing with varying compression rate $\nu$ on the HCP T2w dataset. Note that compressed sensing is more challenging due to the complexity of the forward operator, as evidenced by the poor performance of OT-ODE, which assumes a Gaussian distribution of measurement $y$ given $x_t$. The best values are highlighted in blue.

| Method | $\nu = 1/2$ PSNR | $\nu = 1/2$ SSIM | $\nu = 1/4$ PSNR | $\nu = 1/4$ SSIM | $\nu = 1/10$ PSNR | $\nu = 1/10$ SSIM |
|---|---|---|---|---|---|---|
| Wavelet Prior | $18.02 \pm 1.38$ | $0.495 \pm 0.02$ | $11.99 \pm 1.34$ | $0.230 \pm 0.02$ | $7.37 \pm 1.85$ | $0.090 \pm 0.02$ |
| TV Prior | $25.36 \pm 2.79$ | $0.657 \pm 0.04$ | $18.70 \pm 2.36$ | $0.496 \pm 0.03$ | $14.38 \pm 3.04$ | $0.309 \pm 0.04$ |
| OT-ODE | $18.71 \pm 1.02$ | $0.422 \pm 0.17$ | $18.16 \pm 1.06$ | $0.271 \pm 0.07$ | $12.21 \pm 1.43$ | $0.096 \pm 0.04$ |
| DPS-ODE | $31.06 \pm 3.91$ | $0.765 \pm 0.08$ | $25.01 \pm 1.87$ | $0.608 \pm 0.08$ | $22.06 \pm 1.66$ | $0.479 \pm 0.09$ |
| Ours | $32.72 \pm 1.53$ | $0.878 \pm 0.05$ | $27.03 \pm 1.77$ | $0.733 \pm 0.04$ | $24.03 \pm 1.23$ | $0.503 \pm 0.04$ |

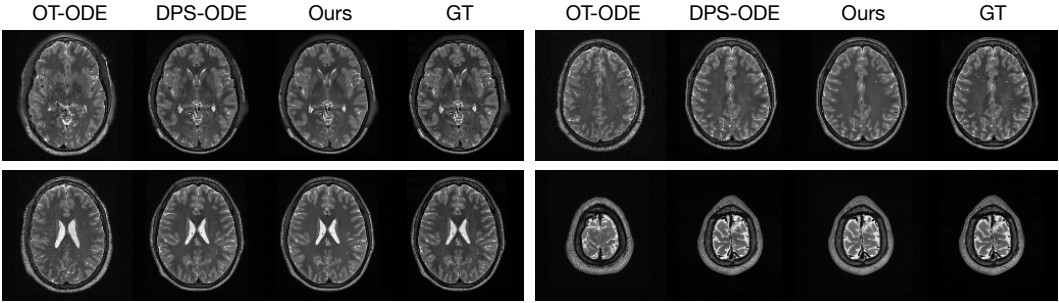

Figure 4: Qualitative comparison results on compressed sensing. Our method produces more faithful reconstructions with fewer artifacts, ensuring higher accuracy and clarity in the details.

In contrast, our method is significantly faster (approximately 1.6 minutes per image) due to our principled local MAP approximation, as demonstrated in Appendix D.

## 4.1 Natural Images

**Experimental setup** We evaluate our algorithm using 100 images from the CelebA-HQ validation set with a resolution of 256×256, normalizing all images to the $[0, 1]$ range for quantitative analysis. All experiments incorporate Gaussian measurement noise with $\sigma_y = 0.01$. We address the following linear inverse problems: (1) 4× super-resolution using bicubic downsampling, (2) inpainting with a random mask covering 70% of missing values, (3) Gaussian deblurring with a 61×61 kernel and a standard deviation of 3.0, and (4) box inpainting with a centered 128×128 mask.

We present the quantitative and qualitative results of all the methods in Tab. 1 and Fig. 3, respectively. In Tab. 1, our method surpasses all other baselines across all tasks. For more challenging tasks such as Gaussian deblurring and box inpainting, our method significantly outperforms others in terms of SSIM. Based on the MAP framework, as shown in Fig. 3, our method prefers more faithful and artifact-free reconstructions, whereas others trade off for perceptual quality. We note that there is an unavoidable tradeoff between perceptual quality and restoration faithfulness [3]. Overall, our method presents a higher degree of refinement. The comparison between ours and ours (w/o prior) indicates the effectiveness of the local prior term in enhancing the accuracy of the reconstructions, as evidenced by the increases in both PSNR and SSIM.

## 4.2 Medical application

**HCP T2w dataset** We utilize images from the publicly available Human Connectome Project (HCP) [50] T2-weighted (T2w) images dataset for the task of compressed sensing, which contains brain images from 47 patients. The HCP dataset includes cross-sectional images of the brain taken at different levels and angles.

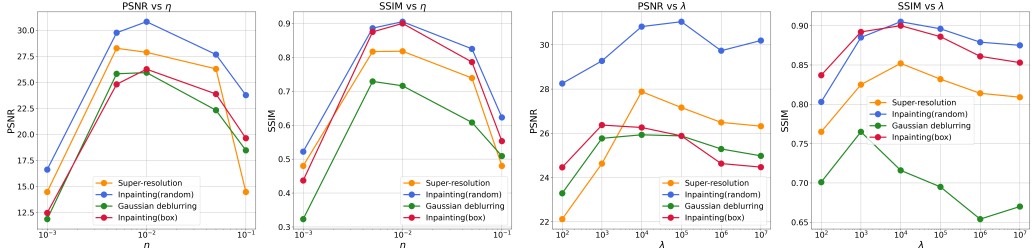

Figure 5: Ablation results of step size $\eta$ and guidance weight $\lambda$. The choice of hyperparameters for our algorithm is fairly consistent across all tasks. We choose $\eta = 10^{-2}$ for all experiments on CelebA-HQ. For $\lambda$, we choose $\lambda = 10^3$ for Gaussian deblurring and $\lambda = 10^4$ for the other tasks.

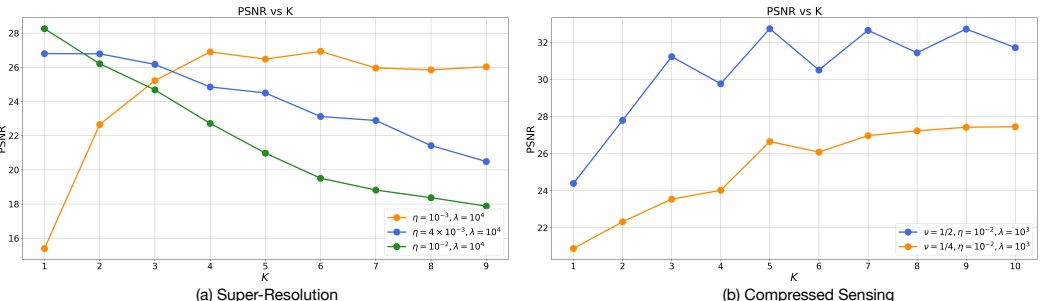

Figure 6: Ablation results of iteration number $K$ on different tasks. For super-resolution and the other three tasks, $K = 1$ is sufficient to achieve the best performance with the optimal step size $\eta$ and guidance weight $\lambda$. However, for compressed sensing, it is necessary to increase $K$ to obtain the best performance. We hypothesize that this is due to the increased complexity of the compressed sensing operator, which requires more iteration steps to ensure the correct optimization direction.

**Compressed sensing**   We train a flow-based model from scratch on 10,000 randomly sampled images, utilizing the *ncsnpp* architecture [48] with minor adaptations for grayscale images. We employ compression rates $\nu \in \{1/2, 1/4, 1/10\}$, meaning $m = \nu n$. The measurement operator is given by a subsampled Fourier matrix, whose sign patterns are randomly selected. We evaluate our reconstruction algorithm's performance on 200 randomly sampled test images.

We present the quantitative and qualitative results of compressed sensing in Tab. 2 and Fig. 4, respectively. In addition to flow-based methods, we include results for two classical recovery algorithms, Wavelet [11, 32] and TV [22] priors. As shown in Tab. 2, our method outperforms the classical recovery algorithms and other flow-based baselines across varying compression rates $\nu$, demonstrating our method's capability to handle challenging scenarios and the advantages of utilizing modern generative models as priors. In Fig. 4, our method produces reconstructions that are more faithful to the original images, with fewer artifacts, leading to higher accuracy and clearer details.

### 4.3   Ablation studies

We use the Adam optimizer [26] for our optimization steps due to its effectiveness in neural network computations. For all tasks, we utilize $N = 100$ steps.

**Step size $\eta$ and Guidance weight $\lambda$**   The use of the Adam optimizer ensures that the choice of hyperparameters, particularly the step size $\eta$ and the guidance weight $\lambda$, remains consistent across various tasks, as illustrated in Fig. 5. Specifically, a step size of $\eta = 10^{-2}$ is optimal for Inpainting (random), Inpainting (box), and Super-resolution in terms of SSIM. For PSNR, Gaussian deblurring also achieves optimal performance at $\eta = 10^{-2}$. Consequently, we employ $\eta = 10^{-2}$ for all tasks. Based on the results shown in the right two subfigures of Fig. 5, we select $\lambda = 10^3$ for Gaussian

deblurring and $\lambda = 10^4$ for the other tasks. This consistency extends to the compressed sensing experiments, where we set $\lambda = 10^3$ and $\eta = 10^{-2}$ for all experiments involving medical images.

**Iteration number** $K$    We present ablation results of the iteration number $K$ on different tasks in Fig. 6. We focus on the behavior of $K$ in super-resolution and compressed sensing, as it performs similarly to super-resolution in the other three tasks. With the optimal choice of $\eta$ and $\lambda$ in super-resolution, i.e., $\eta = 10^{-2}$ and $\lambda = 10^3$, $K = 1$ provides superior performance on CelebA-HQ. A decreased step size, e.g., $\eta = 10^{-3}$, can help performance as $K$ increases, but it fails to exceed the performance achieved with the optimal parameters at $K = 1$. However, for compressed sensing, it is necessary to increase $K$ to achieve the best performance. Consequently, we set $K = 10$ for all compressed sensing experiments. We hypothesize that the complexity of the compressed sensing operator directly determines the number of iterations required for optimal performance.

## 5    Conclusion

In this work, we have introduced a novel iterative algorithm to incorporate flow priors to solve linear inverse problems. By addressing the computational challenges associated with the slow log-likelihood calculations inherent in flow matching models, our approach leverages the decomposition of the MAP objective into multiple "local MAP" objectives. This decomposition, combined with the application of Tweedie's formula, enables effective sequential optimization through gradient steps. Our method has been rigorously validated on both natural and scientific images across various linear inverse problems, including super-resolution, deblurring, inpainting, and compressed sensing. The empirical results indicate that our algorithm consistently outperforms existing techniques based on flow matching, highlighting its potential as a powerful tool for high-resolution image synthesis and related downstream tasks.

## 6    Limitations and Future Work

While our algorithm has demonstrated promising results, there are certain limitations that suggest avenues for future research. First, our theoretical framework, built on optimal transport interpolation paths, is currently limited and cannot be applied to solve the general interpolation between Gaussian and data distributions. Additionally, in order to broaden the applicability of flow priors for inverse problems, it is important to generalize our approach to handle nonlinear forward models. Moreover, the algorithm currently lacks the capability to quantify the uncertainty of the generated images, an aspect crucial for many scientific applications. It would be interesting to consider approaches to post-process our solutions to understand the uncertainty inherent in our reconstruction. These limitations highlight important directions for future work to enhance the robustness and applicability of our method.

## Acknowledgements

The work was partially supported by NSF DMS-2015577, NSF DMS-2415226, and a gift fund from Amazon. We thank anonymous reviewers for their feedback and suggestions, which helped improve the quality of the paper.

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

# Appendix

## A Proof

Before we dive into the proof, we provide the following three lemmas.

**Lemma 1.** *Consider a vector-valued function $f : [0,1] \to \mathbb{R}^n$. Then for any $t \in [0,1]$, we have that*

$$\left\| \int_0^t f(s)ds \right\|^2 \leq \int_0^t \|f(s)\|^2 ds.$$

*Proof.* For each $s \in [0,1]$, let $f_i(s) \in \mathbb{R}$ denote the $i$-th component of $f(s)$. Recall Jensen's inequality: for any convex function $g : \mathbb{R} \to \mathbb{R}$ and integrable function $h : [0,1] \to \mathbb{R}$, we have

$$g\left( \int_a^b h(t)dt \right) \leq \int_a^b g(h(t))dt.$$

Using convexity of the function $t \mapsto t^2$ and applying Jensen's inequality, we see that

$$\begin{aligned}
\left\| \int_0^t f(s)ds \right\|^2 &= \sum_{i=1}^n \left( \int_0^t f_i(s)ds \right)^2 \\
&\leq \sum_{i=1}^n \int_0^t f_i(s)^2 ds \\
&= \int_0^t \sum_{i=1}^n f_i(s)^2 ds \\
&= \int_0^t \|f(s)\|^2 ds.
\end{aligned}$$

$\square$

**Lemma 2** (Tweedie's Formula [12]). *If $\mu \sim g(\cdot)$, $z|\mu \sim \mathcal{N}(\alpha\mu, \sigma^2 I)$, and therefore $z \sim f(\cdot)$, we have*

$$\mathbb{E}[\mu|z] = \frac{1}{\alpha}[z + \sigma^2 \nabla_z \log f(z)]. \tag{12}$$

**Lemma 3.** *Suppose $y = \mathcal{A}(x_*) + \epsilon$ where $x_* = x_1(x_0)$ with $x_0$ being the solution to Eq. (9), $\mathcal{A} : \mathbb{R}^n \to \mathbb{R}^m$ is linear, $\epsilon \sim \mathcal{N}(0, \sigma_y^2 I)$, and $x_t$ exactly follows the path $x_t = \alpha_t x + \beta_t x_0$ for any time $t \in [0,1]$. Then we have*

$$p(y_t|x_t) = \mathcal{N}(\mathcal{A}x_t, \alpha_t^2 \sigma_y^2 I), \tag{13}$$

*and hence*

$$\log p(y|x(x_0)) = \log p(y_t|x_t) + \frac{m}{2}\log(\alpha_t^2), \forall t. \tag{14}$$

*Proof.* Recall that the generated auxiliary path $y_t = \alpha_t y + \beta_t \mathcal{A}x_0$. By assumption, we have $\mathcal{A}(x_t) = \mathcal{A}(\alpha_t x + \beta_t x_0) = \alpha_t \mathcal{A}(x(x_0)) + \beta_t \mathcal{A}x_0$. By subtracting these two equations, we have

$$y_t - \mathcal{A}(x_t) = \alpha_t(y - \mathcal{A}(x(x_0))). \tag{15}$$

As $y|x(x_0) \sim \mathcal{N}(\mathcal{A}x, \sigma_y^2 I)$, we have $y_t|x_t \sim \mathcal{N}(\mathcal{A}x_t, \alpha_t^2 \sigma_y^2 I)$. The proof for Eq. (13) is done. Next, we examine the log probability as follows:

$$\log p(y_t|x_t) = -\frac{\|y_t - \mathcal{A}x_t\|^2}{2\alpha_t^2 \sigma_y^2} - \frac{m}{2}\log(2\pi\alpha_t^2 \sigma_y^2) \tag{16}$$

$$= -\frac{\|\alpha_t(y - \mathcal{A}(x(x_0)))\|^2}{2\alpha_t^2 \sigma_y^2} - \frac{m}{2}\log(2\pi\alpha_t^2 \sigma_y^2) \tag{17}$$

$$= -\frac{\|y - \mathcal{A}(x(x_0))\|^2}{2\sigma_y^2} - \frac{m}{2}\log(2\pi\sigma_y^2) - \frac{m}{2}\log(\alpha_t^2) \tag{18}$$

$$:= \log p(y|x(x_0)) - \frac{m}{2}\log(\alpha_t^2). \tag{19}$$

$\square$

### A.1 Proof of Proposition 1

Trained by the objective defined in Eq. (5), the optimal velocity field would be

$$v_\theta(x_t, t) = \mathbb{E}(\dot{\alpha}_t x_1 + \dot{\beta}_t x_0 | x_t) \tag{20}$$

$$= \mathbb{E}(\dot{\alpha}_t x_1 + \dot{\beta}_t \frac{x_t - \alpha_t x}{\beta_t} | x_t) \qquad \text{\# Given } x_t, x_0 = \frac{x_t - \alpha_t x}{\beta_t} \tag{21}$$

$$= (\dot{\alpha}_t - \dot{\beta}_t \frac{\alpha_t}{\beta_t}) \mathbb{E}(x_1 | x_t) + \frac{\dot{\beta}_t}{\beta_t} x_t \tag{22}$$

$$= (\dot{\alpha}_t - \dot{\beta}_t \frac{\alpha_t}{\beta_t})[\frac{1}{\alpha_t}(x_t + \beta_t^2 \nabla_{x_t} \log p(x_t))] + \frac{\dot{\beta}_t}{\beta_t} x_t. \quad \text{\# Lemma 2(Tweedie's Formula)} \tag{23}$$

By defining the signal-to-noise ratio as $\lambda_t = \alpha_t / \beta_t$ and rearranging the equation above, we get exactly Eq. (11) which we display again below:

$$\nabla_{x_t} \log p(x_t) = \frac{1}{\beta_t^2} \left[ \left( \frac{d \log \lambda_t}{dt} \right)^{-1} \left( v_\theta(x_t, t) - \frac{d \log \beta_t}{dt} x_t \right) - x_t \right]. \tag{24}$$

When $\alpha_t = t$, $\beta_t = 1 - t$, the equation above becomes

$$\nabla_{x_t} \log p(x_t) = \frac{1}{1 - t}(-x_t + t v_\theta(x_t, t)). \tag{25}$$

### A.2 Proof of Theorem 1

Before we dive into the proof, we first point out $\lim_{\Delta t \to 0} \sum_{i=1}^{N} \gamma_i = 1$. Define the timestep $t = (i - 1)\Delta t$. Conversely, $i = 1 + t/\Delta t$ is a function of $t$. In this sense, we define the $i$-th step Riemannian discretization of the integral $-\int_0^1 \text{tr}\left( \frac{\partial v_\theta(x_t, t)}{\partial x} \right) dt$ as $\Delta p_i = -\text{tr}\left( \frac{\partial v_\theta(x_t, t)}{\partial x} \right) \Delta t$.

We first decompose the global MAP objective as follows:

$$\log p(x(x_0)|y) = \log p(x_0) - \int_0^1 \text{tr}\left( \frac{\partial v_\theta(x_t, t)}{\partial x} \right) dt + \log p(y|x(x_0)) - \log p(y) \tag{26}$$

$$= \lim_{\Delta t \to 0} \sum_{i=1}^{N} \gamma_i \log p(x_0) + \lim_{\Delta t \to 0} \sum_{i=1}^{N} \Delta p_i \tag{27}$$

$$+ \lim_{\Delta t \to 0} \sum_{i=1}^{N} \gamma_i [\log p(y_{i\Delta t} | x_{i\Delta t}) + c_i] - \log p(y), \tag{28}$$

where the decomposition of the second term utilizes the property of the discretization of Riemann integral, and that of the third term utilizes the result in Lemma 3 and thus $c_i = \frac{m}{2} \log(\alpha_{i\Delta t}^2)$. By the property of limits, i.e. $\lim_{\Delta t \to 0}(\sum_{i=1}^{N} \gamma_i)(\sum_{i=1}^{N} \Delta p_i) = \lim_{\Delta t \to 0}(\sum_{i=1}^{N} \gamma_i) \lim_{\Delta t \to 0}(\sum_{i=1}^{N} \Delta p_i) = \lim_{\Delta t \to 0} \sum_{i=1}^{N} \Delta p_i$, we can further decompose the second term in Eq. (28) into $\lim_{\Delta t \to 0}(\sum_{i=1}^{N} \gamma_i)(\sum_{i=1}^{N} \Delta p_i)$.

By extracting the limit out in Eq. (28), the equation becomes

$$\lim_{\Delta t \to 0} \Big\{ \gamma_1 \left[ \log p(x_0) + \Delta p_1 + \log p(y_{\Delta t} | x_{\Delta t}) + c_1 \right]$$
$$+ \gamma_2 \left[ \log p(x_0) + \Delta p_1 + \Delta p_2 + \log p(y_{2\Delta t} | x_{2\Delta t}) + c_2 \right]$$
$$+ \cdots$$
$$+ \gamma_N \left[ \log p(x_0) + \Delta p_1 + \Delta p_2 + \cdots + \Delta p_N + \log p(y_{N\Delta t} | x_{N\Delta t}) + c_N \right]$$
$$+ \left[ \gamma_1 \Delta p_2 + (\gamma_1 + \gamma_2) \Delta p_3 + \cdots + (\gamma_1 + \gamma_2 + \cdots + \gamma_{N-1}) \Delta p_N \right] - \log p(y) \Big\} \tag{29}$$

$$:= \lim_{\Delta t \to 0} \left[ \sum_{i=1}^{N} \gamma_i \tilde{\mathcal{J}}_i + \sum_{j=2}^{N} \left( \sum_{i=1}^{j-1} \gamma_i \right) \Delta p_j + \sum_{i=1}^{N} \gamma_i c_i - \log p(y) \right], \tag{30}$$

where $\tilde{\mathcal{J}}_i := \log p(x_0) + \sum_{j=1}^{i} \Delta p_j + \log p(y_{i\Delta t}|x_{i\Delta t})$. We further define the $c(N) = \sum_{i=1}^{N} \gamma_i c_i - \log p(y)$.

Recall that $\hat{\mathcal{J}}_i = \log p(x_{(i-1)\Delta t}) - \text{tr}\left(\frac{\partial v_\theta(x_{(i-1)\Delta t},(i-1)\Delta t)}{\partial x}\right)\Delta t + \log p(y_{i\Delta t}|x_{i\Delta t})$. By triangle inequality, we have

$$\left| \log p(x(x_0)|y) - \sum_{i=1}^{N} \gamma_i \hat{\mathcal{J}}_i - c(N) \right| \tag{31}$$

$$\leqslant \left| \log p(x(x_0)|y) - \sum_{i=1}^{N} \gamma_i \tilde{\mathcal{J}}_i - c(N) \right| + \left| \sum_{i=1}^{N} \gamma_i \hat{\mathcal{J}}_i - \sum_{i=1}^{N} \gamma_i \tilde{\mathcal{J}}_i \right|. \tag{32}$$

Taking the limit on both sides, we have

$$\lim_{\Delta t \to 0} \left| \log p(x(x_0)|y) - \sum_{i=1}^{N} \gamma_i \hat{\mathcal{J}}_i - c(N) \right| \tag{33}$$

$$\leqslant \lim_{\Delta t \to 0} \left| \log p(x(x_0)|y) - \sum_{i=1}^{N} \gamma_i \tilde{\mathcal{J}}_i - c(N) \right| + \lim_{\Delta t \to 0} \left| \sum_{i=1}^{N} \gamma_i \hat{\mathcal{J}}_i - \sum_{i=1}^{N} \gamma_i \tilde{\mathcal{J}}_i \right|. \tag{34}$$

**In the following, we analyze the two terms on the right-hand side one by one. For the first term:** as $|\cdot| : \mathbb{R} \to \mathbb{R}$ is a continuous function, the first term on the right-hand side is equal to

$$\left| \log p(x(x_0)|y) - \lim_{\Delta t \to 0} \sum_{i=1}^{N} \gamma_i \tilde{\mathcal{J}}_i - c(N) \right| \tag{35}$$

$$= \left| \lim_{\Delta t \to 0} \sum_{j=2}^{N} \left( \sum_{i=1}^{j-1} \gamma_i \right) \Delta p_j \right| \tag{36}$$

$$= \left| \lim_{\Delta t \to 0} \sum_{j=2}^{N} \left( \frac{1}{2^{N-j+1}} - \frac{1}{2^N} \right) \Delta p_j \right| \tag{37}$$

$$\leq \left| \lim_{\Delta t \to 0} \sum_{j=2}^{N} \left( \frac{1}{2^{N-j+1}} \right) \Delta p_j \right| + \left| \lim_{\Delta t \to 0} \sum_{j=2}^{N} \left( \frac{1}{2^N} \right) \Delta p_j \right|, \tag{38}$$

where the first equation is derived by subtracting the first term in Eq. (30) from Eq. (26). As the velocity field $v_\theta : \mathbb{R}^n \times \mathbb{R} \to \mathbb{R}^n$ satisfies $\sup_{z \in \mathbb{R}^n, s \in [0,1]} |\text{tr} \frac{\partial}{\partial x} v_\theta(z,s)| \leq C_1$ for some universal constant $C_1$, we have $|\Delta p_j| \leq C_1 \Delta t$. The first term in (38) would be

$$\left| \sum_{j=2}^{N} \left( \frac{1}{2^{N-j+1}} \right) \Delta p_j \right| \leq C_1 \Delta t \sum_{j=2}^{N} \left( \frac{1}{2^{N-j+1}} \right) \leq C_1 \Delta t = O(\Delta t). \tag{39}$$

Similarly, the second term in (38) would be

$$\left| \sum_{j=2}^{n} \left( \frac{1}{2^n} \right) \Delta p_j \right| \leq \sum_{j=2}^{N} \left( \frac{1}{2^N} \right) C_1 \Delta t = C_1 \left( \frac{N-1}{2^N} \right) \Delta t = O(\Delta t). \tag{40}$$

Combining the results in Eq. (39) and Eq. (40), we get

$$\left| \log p(x(x_0)|y) - \lim_{\Delta t \to 0} \sum_{i=1}^{N} \gamma_i \tilde{\mathcal{J}}_i - c(N) \right| = 0. \tag{41}$$

**For the second term:** Intuitively, the error between the integral and the Riemannian discretization goes to 0 as $\Delta t$ tends to 0. Rigorously,

$$\lim_{\Delta t \to 0} \left| \sum_{i=1}^{N} \gamma_i \hat{\mathcal{J}}_i - \sum_{i=1}^{N} \gamma_i \tilde{\mathcal{J}}_i \right| = \lim_{\Delta t \to 0} \left| \sum_{i=1}^{N} \gamma_i (\hat{\mathcal{J}}_i - \tilde{\mathcal{J}}_i) \right| \tag{42}$$

$$= \lim_{\Delta t \to 0} \left| \sum_{i=1}^{N} \gamma_i \left( \int_0^{t-\Delta t} \text{tr} \left( \frac{\partial v_\theta(x_s, s)}{\partial x} \right) ds - \sum_{j=1}^{i-1} \Delta p_j \right) \right| \tag{43}$$

$$\leq \lim_{\Delta t \to 0} \sum_{i=1}^{N} \gamma_i \left| \int_0^{t-\Delta t} \text{tr} \left( \frac{\partial v_\theta(x_s, s)}{\partial x} \right) ds - \sum_{j=1}^{i-1} \Delta p_j \right| = 0. \tag{44}$$

**Combining the results of the first term and the second term, we get the proof of theorem 1 done.**

## B  Compliance of Trajectory

To quantify our deviation from the assumption of having $x_t$ exactly follow the interpolation path $\alpha_t x + \beta_t x_0$, we define the following: given a differentiable process $\{z_t\}$ and an interpolation path specified by $\boldsymbol{\alpha} := \{\alpha_t\}$ and $\boldsymbol{\beta} := \{\beta_t\}$, we define the trajectory's **compliance** $S_{\boldsymbol{\alpha}, \boldsymbol{\beta}}(\{z_t\})$ to the interpolation path as

$$S_{\boldsymbol{\alpha}, \boldsymbol{\beta}}(\{z_t\}) := \int_0^1 \mathbb{E}_{p(z_0), p(z_1)} \left[ \|\dot{z}_t - (\dot{\alpha}_t z_1 + \dot{\beta}_t z_0)\|^2 \right] dt. \tag{45}$$

This generalizes the definition of straightness in [29] to general interpolation paths. We recover their definition by setting $\alpha_t = t$ and $\beta_t = 1 - t$. In certain cases, we have exact compliance with the predefined interpolation path. For example, when $\{z_t\}$ is generated by $v_\theta$ and $\alpha_t = t$ and $\beta_t = 1 - t$, note that $S_{\boldsymbol{\alpha}, \boldsymbol{\beta}}(\{z_t\}) = 0$ is equivalent to $v_\theta(z_t, t) = c$ where $c$ is a constant, almost everywhere. This ensures that $z_1 = z_0 + c$. In this case, when generating the trajectory through an ODE solver with starting point $x_0$ and endpoint $x_t$, we have $x_t = \alpha_t x + \beta_t x_0, \forall t$. When $S_{\boldsymbol{\alpha}, \boldsymbol{\beta}}(\{z_t\})$ is not equal to 0, we show in Proposition 2 that we can bound the deviation of our trajectory from the interpolation path using this compliance measure. When specifying our result to Rectified Flow, we can obtain an additional bound showing that when using $L$-Rectified Flow, the deviation of the learned trajectory from the straight trajectory is bounded by $O(1/L)$.

**Proposition 2.** *Consider a differentiable interpolation path specified by $\boldsymbol{\alpha} := \{\alpha_t\}$ and $\boldsymbol{\beta} := \{\beta_t\}$. Then the expected distance between the learned trajectory $z_t = z_0 + \int_0^t v_\theta(z_s, s) ds$ and the predefined trajectory $\hat{z}_t = z_0 + \int_0^t (\dot{\alpha}_s z_1 + \dot{\beta}_s z_0) ds$ can be bounded as*

$$\mathbb{E}_{p(z_0), p(z_1)} \left[ \|\hat{z}_t - z_t\|^2 \right] \leq S_{\boldsymbol{\alpha}, \boldsymbol{\beta}}(\{z_t\}). \tag{46}$$

*If the differentiable process $\{z_t\}$ is specified by L-Rectified Flow and $\alpha_t = t$ and $\beta_t = 1 - t$ for all $t \in [0, 1]$, then we additionally have*

$$\mathbb{E}_{p(z_0), p(z_1)} \left[ \|\hat{z}_t - z_t\|^2 \right] \leq O \left( \frac{1}{L} \right). \tag{47}$$

*Proof.* At time $t$, we are interested in the distance between a real trajectory $z_t = z_0 + \int_0^t v_\theta(z_s, s) ds$ and a preferred trajectory $\hat{z}_t = z_0 + \int_0^t (\dot{\alpha}_s z_1 - \dot{\beta}_s z_0) ds$. Using the result in Lemma 1, the distance can be bounded by

$$\|\hat{z}_t - z_t\|^2 = \left\| \int_0^t [v_\theta(z_s, s) - (\dot{\alpha}_s z_1 - \dot{\beta}_s z_0)] ds \right\|^2 \tag{48}$$

$$\leq \int_0^t \|v_\theta(z_s, s) - (\dot{\alpha}_s z_1 - \dot{\beta}_s z_0)\|^2 ds. \tag{49}$$

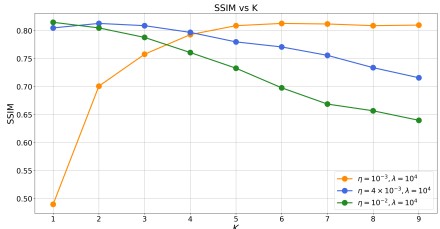 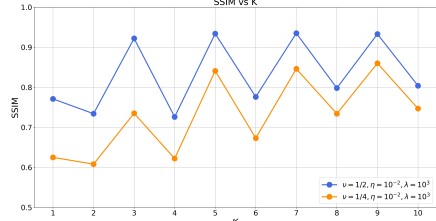

Figure 7: Ablation results of $K$ in terms of SSIM on different tasks.

Therefore,

$$\mathbb{E}_{p(z_0),p(z_1)}\|\hat{z}_t - z_t\|^2 \leq \mathbb{E}_{p(z_0),p(z_1)}\left[\int_0^t \|v_\theta(z_s,s) - (\dot{\alpha}_s z_1 - \dot{\beta}_s z_0)\|^2 ds\right] \tag{50}$$

$$= \int_0^t \mathbb{E}_{p(z_0),p(z_1)}\|v_\theta(z_s,s) - (\dot{\alpha}_s z_1 - \dot{\beta}_s z_0)\|^2 ds \tag{51}$$

$$\leq \int_0^1 \mathbb{E}_{p(z_0),p(z_1)}\|v_\theta(z_s,s) - (\dot{\alpha}_s z_1 - \dot{\beta}_s z_0)\|^2 ds \tag{52}$$

$$:= S_{\boldsymbol{\alpha},\boldsymbol{\beta}}(\{z\}). \tag{53}$$

If $\{z_t, t \in [0,1]\}$ is a learned $L$-rectfied flow, i.e. $\alpha_t = t$ and $\beta_t = 1 - t$ in this case, where $L$ is the times of rectifying the flow, by Theorem 3.7 in [29], we have $S_{\boldsymbol{\alpha},\boldsymbol{\beta}}(\{z\}) = O(1/L)$ and thus

$$\mathbb{E}_{p(z_0),p(z_1)}\|\hat{z}_t - z_t\|^2 = O(1/L). \tag{54}$$

$\square$

Empirically, [30, 29] found $L = 2$ generates nearly straight trajectories for high-quality one-step generation. Hence, while this result gives us a simple upper bound, in practice the trajectories may comply more faithfully with the predefined interpolation path than this result suggests.

## C Additional Results

### C.1 Additional Ablations

**Iteration steps $K$** We provide additional ablation results of $K$ in terms of SSIM in Fig. 7.

**NFEs $N$** We first refer to Fig. 2(c) for a preliminary ablation on $N$ using a toy example. Next, we show PSNR and SSIM scores for varying $N$ in the task of super-resolution. We find that $N = 100$ is the best trade-off between time and performance. The ablation results are shown in Fig. 8.

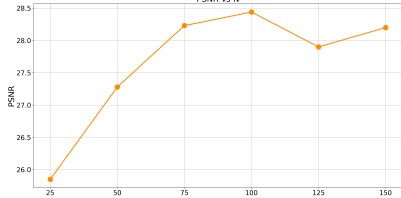 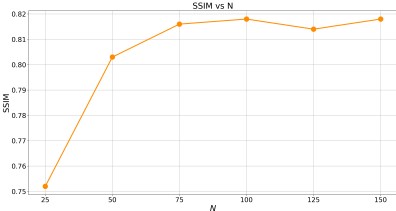

Figure 8: Ablation results of the NFEs $N$ on the super-resolution task.

## D Computational Efficiency

In Tab. 3, we present the computational efficiency comparison results. Note that OT-ODE is the slowest as it requires taking the inverse of a matrix $r_t^2 \mathcal{A}\mathcal{A}^T + \sigma_y^2 I$ each update time. Our method requires taking the gradient over an estimated trace of the Jacobian matrix, which slows the computation.

Table 3: **Computational time comparison.** We compare the time required to recover 100 images for the super-resolution task on a single GPU.

|  | DPS-ODE | OT-ODE | Ours (w/o prior) | Ours |
|---|---|---|---|---|
| Time(h) | 0.36 | 4.10 | 0.83 | 2.72 |

# E   Implementation Details

Experiments were conducted on a Linux-based system with CUDA 12.2 equipped with 4 Nvidia R9000 GPUs, each of them has 48GB of memory.

**Operators**   For all the experiments on the CelebA-HQ dataset, we use the operators from [10]. For all the experiments on compressed sensing, we use the operator *CompressedSensingOperator* defined in the official repository of [14] [4],

**Evaluation**   Metrics are implemented with different Python packages. PSNR is calculated using basic PyTorch operations, and SSIM is computed using the *pytorch_msssim* package.

## E.1   Toy example

The workflow begins with using 1,000 FFHQ images at a resolution of $1024\times1024$. These images are then downscaled to $16\times16$ using bicubic resizing. A Gaussian Mixture model is applied to fit the downsampled images, resulting in mean and covariance parameters. The mean values are transformed from the original range of [0,1] to [-1,1]. Subsequently, 10,000 samples are generated from this distribution to facilitate training a score-based model resembling the architecture of CIFAR10 DDPM++. The training process involves 10,000 iterations, each with a batch size of 64, and utilizes the Adam optimizer [26] with a learning rate of 2e-4 and a warmup phase lasting 100 steps. Notably, convergence is achieved within approximately 200 steps. Lastly, the estimated log-likelihood computation for a batch size of 128 takes around 4 minutes and 30 seconds. We show uncured samples generated from the trained models in Fig. 9.

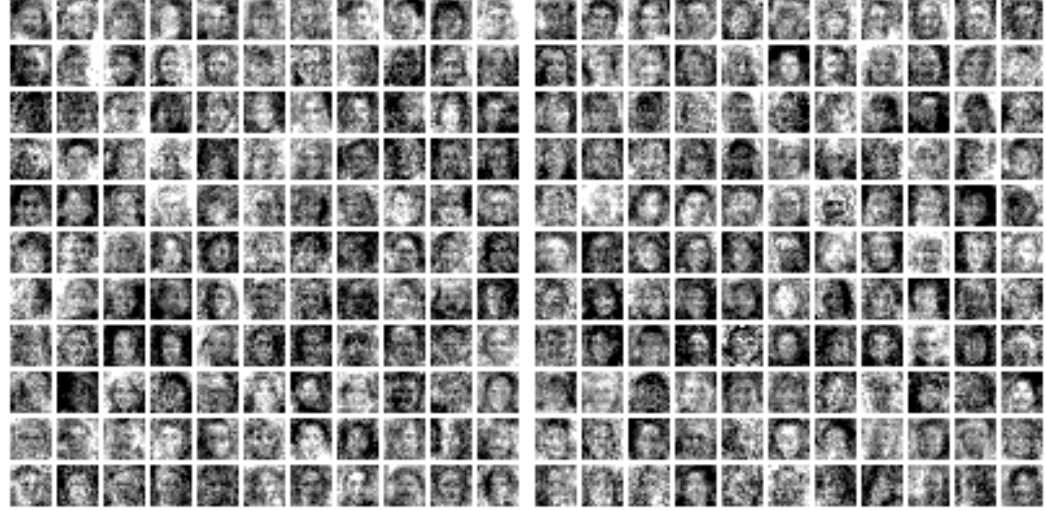

Figure 9: Generated samples from the flow trained on 10,000 Gaussian samples.

## E.2   Medical Application

In this setting, $\sigma_y = 0.001$. We use the *ncsnpp* architecture, training from scratch on 10k images for 100k iterations with a batch size of 50. We set the learning rate to $1 \times 10^{-2}$. Sudden convergence appeared during our training process. We use 2000 warmup steps. Uncured generated images are presented in Fig. 10.

---

[4] https://github.com/Sulam-Group/learned-proximal-networks/tree/main

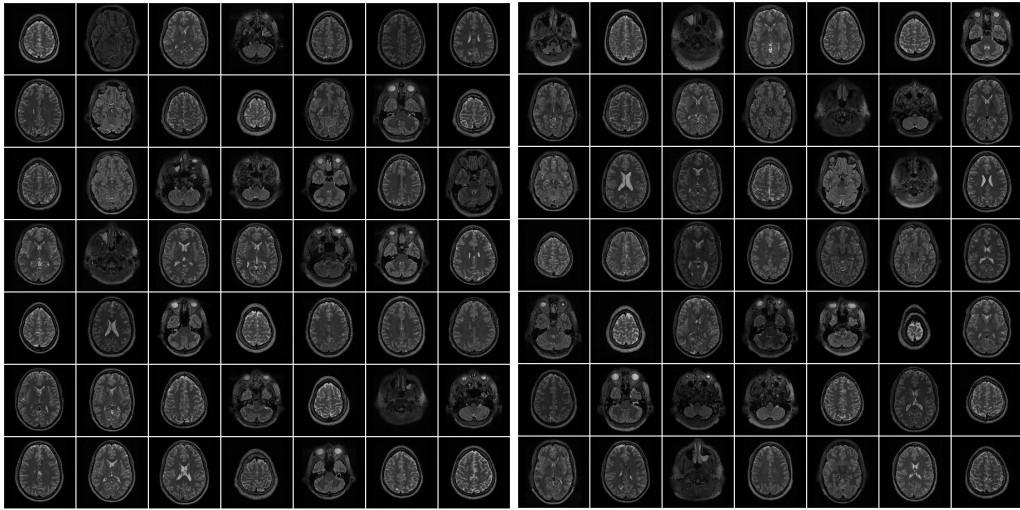

Figure 10: Generated samples from the flow trained on 10,000 HCP T2w images.

### E.3 Implementation of Baselines

**OT-ODE**  As OT-ODE [37] has not released their code and pretrained checkpoints. We reproduce their method with the same architecture as in [29]. We follow their setting and find initialization time $t'$ has a great impact on the performance. We use the *y init* method in their paper. Specifically, the starting point is

$$x_{t'} = t'y + (1 - t')\epsilon, \ \epsilon \sim \mathcal{N}(0, I), \tag{55}$$

where $t'$ is the init time. Note that in the super-resolution task we upscale $y$ with bicubic first. We follow the guidance in the paper and show the ablation results in Fig. 11 and Fig. 12.

Super-Resolution      Inpainting(random)      Gaussian Deblurring      Inpainting(box)

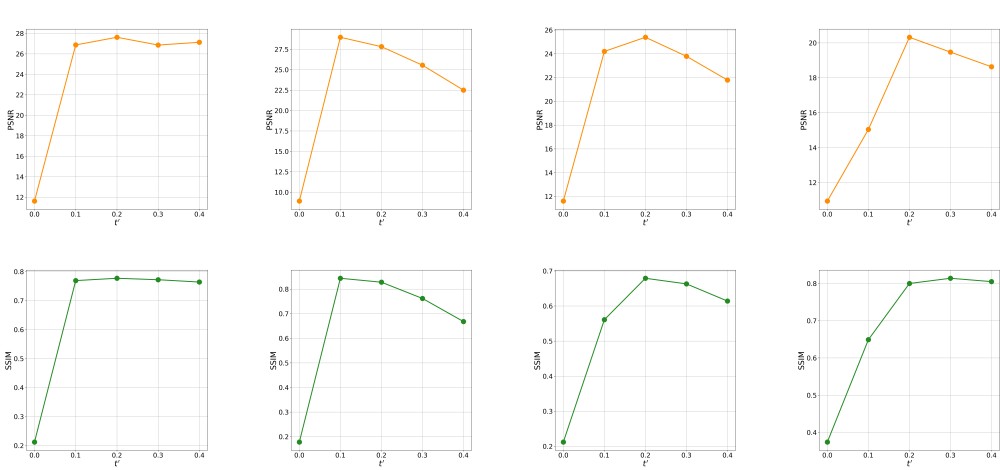

Figure 11: Hyperparameter $t'$ selection results for OT-ODE on the CelebA-HQ dataset. We select $t' = 0.2, 0.1, 0.2, 0.2$ for super-resolution, inpainting(random), Gaussian deblurring, and inpainting(box), respectively.

**DPS-ODE**  We use the following formula to update for each step in the flow:

$$v(x_t, y) = v(x_t) + \zeta_t \left( -\nabla_{x_t} \|y - \mathcal{A}\hat{x}_1\|^2 \right),$$

where $\zeta_t$ is the step size to tune. We refer to DPS for the method to choose $\zeta_t$. We set $\zeta_t = \frac{\eta}{2\|y - \mathcal{A}\hat{x}_1(x_t)\|}$. We demonstrate the ablation of $\eta$ for this baseline in Fig. 13 and Fig. 14. Note that there is a significant divergence in PSNR and SSIM for the task of inpainting (box). As we observe that artifacts are likely to appear when $\eta \geq 100$, we choose the optimal $\eta = 75$ for the best tradeoff.

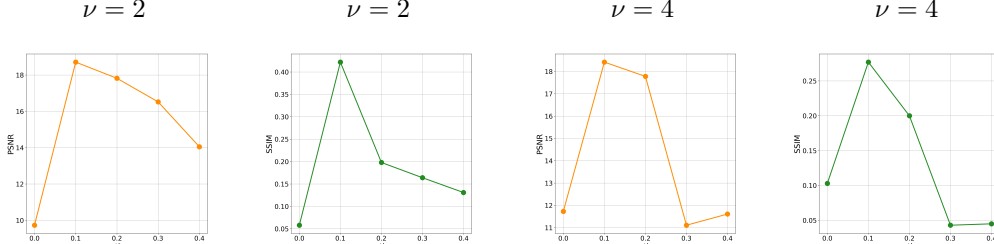

Figure 12: Hyperparameter $t'$ selection results for OT-ODE on the HCP T2w dataset. We select $t' = 0.1$ for all the experiments.

**RED-Diff and $\Pi$GDM**  We use the official repository[5] from Nvidia to reproduce the results of RED-Diff and $\Pi$GDM with the pretrained CelebAHQ checkpoint using the architecture of the guided diffusion repository[6] from OpenAI.

**For RED-Diff**, the optimization objective is $\min_\mu ||y - \mathcal{A}(\mu)||^2 + \lambda(sg(\epsilon_\theta(x_t, t) - \epsilon))^T\mu$. Following the implementation of the original paper, we use Adam optimizer with 1,000 steps for all tasks. We choose learning rate $lr = 0.25, \lambda = 0.25$ for super-resolution, inpainting(random) and inpainting(box) and $lr = 0.5, \lambda = 0.25$ for deblurring as recommended by the paper.

**For $\Pi$GDM**, we follow the original paper and use 100 diffusion steps. Specifically, we use $\eta = 1.0$ which corresponds to the VE-SDE. Adaptive weights $r_t^2 = \frac{\sigma_{1-t}^2}{1+\sigma_t^2}$ are used if there is an improvement on metrics.

**Wavelet and TV priors**  We use the pytorch package DeepInverse[7] to implement Wavelet and TV priors. For both priors, we use the default Proximal Gradient Descent (PGD) algorithm and perform a grid search for regularization weight $\lambda$ in the set $\{10^0, 10^{-1}, 10^{-2}, 10^{-3}, 10^{-4}\}$ and gradient stepsize $\eta$ in $\{10^1, 10^0, 10^{-1}, 10^{-2}, 10^{-3}, 10^{-4}\}$. The maximum number of iteration is 3k, 5k, and 10k for compression rate $\nu = 1/2, 1/4$, and $1/10$, respectively. The stopping criterion is the residual norm $\frac{||x_{t-1} - x_t||}{||x_{t-1}||} \leq 1 \times 10^{-5}$ and the initialization of the algorithm is the backprojected reconstruction, i.e., the pseudoinverse of $\mathcal{A}$ applied to the measurement $y$.

**For the TV prior**, the objective we aim to minimize is $\min_x \frac{1}{2}||\mathcal{A}x - y||_2^2 + \lambda||x||_{TV}$. We find that the optimal combination of hyperparameters is $\lambda = 0.01, \eta = 0.1$ for all the values of $\nu$.

**For the Wavelet prior**, the objective we want to minimize is $\min_x \frac{1}{2}||\mathcal{A}x - y||_2^2 + \lambda||\Psi x||_1$. We use the default level of the wavelet transform and select the "db8" Wavelet. The optimal combination of hyperparameters is $\lambda = 0.1, \eta = 0.1$ for all the values of $\nu$.

---

[5]https://github.com/NVlabs/RED-diff
[6]https://github.com/openai/guided-diffusion
[7]https://deepinv.github.io/deepinv/

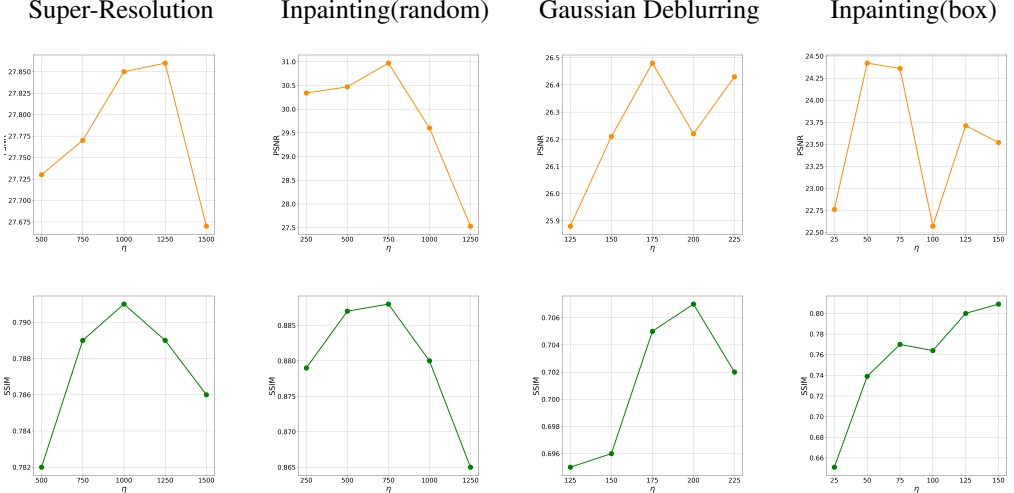

Figure 13: Hyperparameter $\eta$ selection results for DPS-ODE. We select $\eta = 1000, 750, 200, 75$ for super-resolution, inpainting(random), Gaussian deblurring, and inpainting(box), respectively.

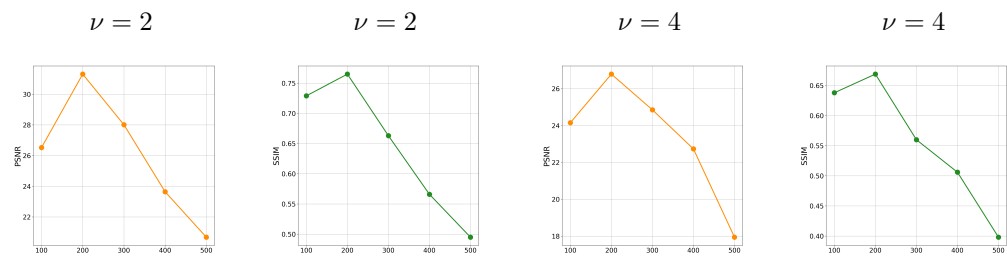

Figure 14: Hyperparameter $\eta$ selection results for DPS-ODE on the HCP T2w dataset. We select $\eta = 200$ for all the experiments.

