# OpenReview forum: "Flow Priors for Linear Inverse Problems via  Iterative Corrupted Trajectory Matching"
_NeurIPS.cc/2024/Conference — NeurIPS 2024 poster_

### Official Review · Reviewer_qjx2 · 2024-07-11

**Soundness:** 3
**Presentation:** 2
**Contribution:** 2
**Rating:** 5
**Confidence:** 4

**Summary:**

The paper investigates the application of flow matching-based generative models for high-resolution image synthesis, particularly as priors for solving inverse problems. A notable challenge addressed is the slow computation of log-likelihoods in high-dimensional contexts, necessitating backpropagation through an ODE solver. To overcome this, the authors propose an efficient iterative algorithm for approximating the maximum-a-posteriori (MAP) estimator. This method involves approximating the MAP objective through a series of "local MAP" objectives and employs Tweedie's formula for sequential gradient optimization. The proposed method's performance is validated across multiple inverse problems and various datasets.

**Strengths:**

- The paper introduces a unique method for solving inverse problems and reconstructing a single image.
- The proposed method is supported by a solid theoretical foundation.
- The approach is effective across diverse linear inverse problems.

**Weaknesses:**

- Insufficient Empirical Evidence: The paper lacks enough qualitative results and empirical evidence to conclusively demonstrate the superiority of the proposed method.
- Need for Additional Metrics: Metrics like FID and LPIPS scores should be included alongside PSNR and SSIM.
- Figure 2 Presentation: The data in Figure 2 should be presented in a table format for easier interpretation of quantitative performance improvements.
- Limited Dataset Testing: For natural image experiments, the algorithm was only tested on the CelebA-HQ dataset. Pretrained models for other datasets (LSUN-bedroom, LSUN-church, and AFHQ-cat) are publicly available from the authors of the rectified flow paper and should be included.
- Blurry reconstruction results:  A closer look at Figure 3 (a,c) reveals that the reconstructed images are too smooth and lack high-frequency details. Compared to OT-ODE, the results are smoother and blurrier.
- Comparison with Recent Work: Apart from OT-ODE, recent work like "D-Flow: Differentiating through Flows for Controlled Generation" also addresses inverse problems and should be considered for comparison.

**Questions:**

- Does the proposed method work for very noisy corruptions with high measurement noise (e.g., sigma_y = 0.2 or more)?
- Does the proposed method work for severely ill-posed inverse problems like 8x super-resolution or inpainting with 90% of pixels missing?
- Why was the algorithm evaluated on only 100 images from the CelebA-HQ dataset? Why wasn't the entire validation dataset used?
-There are numerous qualitative results for medical images in the appendix, but for natural image datasets like CelebA-HQ, there is only one small figure (Figure 3). It is hard to draw any conclusions based on just Figure 3. Was the authors' focus primarily on medical images?

**Limitations:**

- The paper's empirical validation is limited, requiring more comprehensive testing across various datasets and additional qualitative results to strengthen the claims of superiority.
- The presentation of quantitative results could be improved by using tables for easier comparison and interpretation.
- The evaluation metrics need to be expanded to include FID and LPIPS scores to provide a more comprehensive assessment of performance.
- Blurry results

---

> ### Author Rebuttal · Authors · 2024-08-06
>
> We thank you for your helpful review and provide the additional experiments which we hope address your concerns:
>
> **More baselines: RED-Diff, $\Pi$GDM, and D-Flow** See Tab. 2 in the attached pdf above. In addition to the flow-based baselines, we have included two representative diffusion-based baselines: 1) RED-Diff [1], a variational Bayes-based method; and 2) $\Pi$GDM [2], an advanced MCMC-based method. Our method demonstrates competitive performance with both diffusion-based baselines and other approaches in terms of recovery fidelity.
>
> Thank you for bringing D-Flow [3], recently accepted to ICML 2024, to our attention. We did not compare with D-Flow as we consider it concurrent work and it has not yet released its official implementation. Notably, D-Flow also includes [1] and [2] as diffusion-based baselines. Implementation details are included at the end of the [Author Rebuttal] above.
>
> **Runtime comparison with D-Flow** As shown in the Table 3 in the appendix of our paper, our method requires 1.6 min for each image, OT-ODE [2] requires 2.46 min, while the concurrent work D-Flow [1], which formulates the MAP into a constrained optimization problem (Eq. 9), requires 5-10 mins for each image as documented in their Sec. 3.4. This is because each of its optimization steps requires a backpropogation through an ODE solver, as they calculate the log-likelihood fully. Our work is significantly faster due to our principled local MAP approximation.
>
> **Highly noisy setting**  See Tab. 2 of the attached pdf above. We present the results for $\sigma_y = 0.2$ along with a qualitative comparison at the end of the document. We adopted the same hyperparameter selection strategy as described in the reference paper. With a fixed step size of $\eta = 10^{-2}$, we observed that decreasing the guidance scale is beneficial in very noisy conditions. Specifically, we chose $\lambda = 10^{2}$ for super-resolution, random inpainting, and Gaussian deblurring, while $\lambda = 10^{1}$ was selected for box inpainting. Our method outperforms other flow-based baselines, demonstrating its robustness in handling very noisy scenarios.
>
> **Severely ill-posed problems** See Tab. 1 of the attached pdf above. We consider increasing the level of ill-posedness in the compressed sensing MRI experiments, as this is a more challenging inverse problem. We set the compression rate to be $\nu=1/10$, i.e. only 10\% of the output signal has been observed, which relates to inpainting with 90\% of pixels missing you mentioned. Our method outperforms the classical recovery algorithms and other baselines in all settings, demonstrating our method's capability to handle challenging scenarios and the advantages of utilizing modern generative models as priors. Implementation details are included at the end of the [Author Rebuttal] above.
>
> **LPIPS FID scores** They are provided in Tab. 3 of the attached pdf above. Note that in papers oriented by posterior sampling, more emphasis is put on perceptual quality metrics, such as FID and LPIPS. A large number of images (usually 1k) is thus required due to FID's Gaussian assumptions on the distribution. However, our paper focuses mainly on metrics of recovery fidelity such as PSNR and SSIM, which can be calculated by individual image pairs (the same applies to LPIPS). We find both metrics remain stable after the number of test images reach 100. Note that our method is also competitive with other baselines in terms of perceptual quality as evidenced by our LPIPS scores. The FID score calculated by 100 images is quite noisy and just for reference.
>
>
> We hope our additional experiments on severely ill-posed inverse problems, in highly noisy setting, with more baselines compared and perceptual metrics reported can address some of your concerns. Finally, we want to highlight our theoretical contribution to inverse problems with flow-based models as priors, where backpropagation through an ODE solver (usually requiring 100 or more NFEs) is avoided, and asymptotic convergence is proven, as demonstrated in Theorem 1.
> We kindly request a re-evaluation of the rating if some of your concerns have been resolved.
>
>
>
>
>
>
> [1] A Variational Perspective on Solving Inverse Problems with Diffusion Models. Mardani, Morteza and Song, Jiaming and Kautz, Jan and Vahdat, Arash. ICLR 2024.
>
> [2] Pseudoinverse-guided diffusion models for inverse problems. Song, Jiaming and Vahdat, Arash and Mardani, Morteza and Kautz, Jan. ICLR 2023.
>
> [3] D-Flow: Differentiating through Flows for Controlled Generation. Heli Ben-Hamu, Omri Puny, Itai Gat, Brian Karrer, Uriel Singer, Yaron Lipman. ICML 2024 (concurrent work)

---

> ### Author Response · Authors · 2024-08-12
>
> Dear reviewer,
>
> Thanks a lot for your time and effort. As the discussion period approaches its deadline, we would greatly appreciate your feedback on whether our rebuttal has adequately addressed your concerns. Please reach out if you have additional questions or comments which we are happy to address. Thank you again.
>
> Best, Authors

---

> > ### Comment · Reviewer_qjx2 · 2024-08-12
> >
> > The authors have addressed my concerns by providing empirical evidence for the requested cases. Hence, I raise my score to 5.

---

> > > ### Author Response · Authors · 2024-08-13
> > >
> > > We are glad  to hear that your concerns have been addressed. Thank you for your time and support.

---

### Official Review · Reviewer_sT2G · 2024-07-12

**Soundness:** 3
**Presentation:** 3
**Contribution:** 3
**Rating:** 5
**Confidence:** 4

**Summary:**

The paper proposes a flow prior under the MAP structure for solving inverse problems, a theoretical analysis is also given.

**Strengths:**

1. The paper is easy to follow, the presentation from concepts to methods is concise, the motivation of the proposed method on overcoming shortage of flow model is also clear and straightforward.
2. Incorporating the flow prior into MAP to solve inverse problems is interesting and worth exploring.
3. The discussion on feasibility of assumptions made in the paper, as well as hyperparameter effects are very informative to the readers, and are valuable to the research in the community.

**Weaknesses:**

1. Theorem 1 is trivial in terms of $N \rightarrow 0$, and the existence of constant $c(N)$. The approximation gap converges to 0 as N goes to infinity does not provide insightful guidance on the practical implementation since $N \rightarrow 0$ is infeasible (decreases efficiency) in practice. Also, the implicit representation of approximation error constant $c(N)$ does not make sense because it gives no information about how the error could be scaled with $N$, the authors are expected to provide explicit expression of $c( \cdot )$ as a function of $N$.

2. Obtaining measurements $u_t$ in the ‘corrupted trajectory’ and generating auxiliary path $s_t$ do not make too much sense to me, especially when $x_t$ is very noisy and the forward operator $A(\cdot)$ is challenging. Suppose $A$ is very ill-posed (e.g. compressed sensing with low sampling rate), then the $y_t$ in the auxiliary path is not good itself, nevertheless the measurement $u_t = A(x_t)$ obtained in the corrupted trajectory especially when $x_t$ has high noise level. The assumption on exact compliance of trajectory (between corrupted trajectory and auxiliary path) is not theoretically guaranteed as stated by the authors (line 170-171), and the strong empirical results can not support the existence of the assumption made. The good empirical results could only be valid in less challenging inverse problems, or those inverse problems that are suitable for generative models to solve, e.g. inpainting, super-resolution.

3. In the compressed sensing experiment, the sampling rate $\nu$ is relatively large (0.25, 0.5). The authors are expected to use smaller sampling rates (e.g. 0.05, 0.1), and at the same time compare it with classical recovery algorithms, developed from the seminal work by Donoho, et al [1,2] (no neural networks involved), as well as compressed sensing with other generative models (VAE, GAN, Diffusion). The current results and comparison are not convincing.

[1] Donoho, David L. "Compressed sensing." IEEE Transactions on information theory 52.4 (2006): 1289-1306.

[2] Lustig, Michael, et al. "Compressed sensing MRI." IEEE signal processing magazine 25.2 (2008): 72-82.

4. The $\lambda_t$ used in Proposition 1 as SNR can also be found in previous work [1], the authors are expected to provide some comparison or discussion with [1] although it is based on Diffusion, it uses MAP structure and solve optimization problem during the sampling process.

[1] Mardani, Morteza, et al. "A variational perspective on solving inverse problems with diffusion models." arXiv preprint arXiv:2305.04391 (2023).

5. The number of iterations $K$ affects the performance a lot on different tasks, the authors are expected to provide more explanation on the choice of $K$. Also, how are the $K$ and $\lambda$ picked in the experiments? Given that the testing performances are sensitive to these hyperparameters.

6. The baseline methods compared are limited, only 3 flow based models are picked, generative models have been widely used for image restoration tasks, the authors are expected to provide at least several other representative generative models for comparison.

**Questions:**

N/A

---

> ### Author Rebuttal · Authors · 2024-08-06
>
> We sincerely thank you for your helpful and detailed feedback, which significantly helped us improve the paper quality.
>
> **Dependence on $N$ in Theorem 1** We appreciate your concern regarding our theory. While we agree that it could be beneficial to obtain a non-asymptotic error bound in terms of $N$, the goal of this Theorem is to provide intuition for why our *local MAP* objectives are a reasonable approximation to the global MAP problem. The asymptotic nature of the Theorem simplifies its presentation, while still providing support for why our approach works. The terms that actually govern how the error scales with $N$ are the $\hat{\mathcal{J}}_i$ terms. Obtaining non-asymptotic error bounds in $N$ is possible, but this would require further technical assumptions on the flow model to control the approximation error via Riemannian sums.
>
> The $c(N)$ term is given by $c(N) = \sum_{i=1}^N \gamma_i c_i - \log p(y) =  \sum_{i=1}^N (\frac{1}{2})^{N-i+1} m\log (\frac{i}{N} ) - \log p(y)$. This quantity converges to a constant as $N \rightarrow \infty$. We promise to include the explicit expression of $c(N)$ into Theorem 1 in the revision.
>
>
> **Classical recovery algorithms: Wavelet and TV priors** Thank you for this suggestion. See Tab. 1 in the attached pdf above.  We have added results for two classical recovery algorithms, Wavelet and TV priors. Based on the first two rows, the TV prior consistently surpasses the Wavelet prior. We surmise that this is because the TV prior is more effective for images with clear boundaries and homogeneous regions, such as those in the HCP T2w dataset. We find that our method outperforms the classical recovery algorithms in all settings.
> Implementation details are included at the end of the [Author Rebuttal] above.
>
> **More challenging setting: smaller compression rate $\nu=1/10$** Thank you for this suggestion. See Tab. 1 in the attached pdf above. We conduct an additional experiment using the compression ratio $\nu = 1/10$, i.e only 10\% of the output signal has been observed. Based on the two rightmost columns of Tab.1, we see that our method consistently outperforms all the other baselines in terms of PSNR and SSIM.
>
> **More baselines: RED-Diff and $\Pi$GDM** See Tab. 2 in the attached pdf above. In addition to the flow-based baselines, we have included two representative diffusion-based baselines: 1) RED-Diff [1], a variational Bayes-based method; and 2) $\Pi$GDM [2], an advanced MCMC-based method. Our method demonstrates competitive performance with both diffusion-based baselines and other approaches in terms of recovery fidelity. Additionally, one reviewer mentioned a concurrent work, D-Flow [3], recently accepted to ICML 2024, which has not yet released its official code. Notably, D-Flow also includes [1] and [2] as diffusion-based baselines. Implementation details are included at the end of the [Author Rebuttal] above.
>
> **Hyperparameter tuning for $\lambda, \eta, K$** In our implementation, we found that with the Adam optimizer, the choice of step size $\eta$ and guidance weight $\lambda$ remains consistent. Practically, for a new task, we recommend starting by fixing the iteration number $K$ at 1. An initial step size of $\eta=10^{-2}$ is likely to perform well, as this value was effective across all tasks, from natural images to medical applications, after a comprehensive grid search (refer to Fig 5 in the paper).
> The value of $\lambda$ should then be determined by a grid search within the set {$10^2, 10^3, ..., 10^7$}. After establishing $\lambda$, we suggest incrementally increasing $K$ to observe if performance improves. If performance continues to increase, keep raising $K$ until the metrics plateau; if there is no improvement, $K=1$ is the optimal choice.
>
> We hope that our additional experiments in more challenging setting, with classical recovery algorithms and representative diffusion-based methods compared can address some of your concerns. We kindly request a re-evaluation of the rating if some of your concerns have been resolved.
>
>
> [1] A Variational Perspective on Solving Inverse Problems with Diffusion Models. Mardani, Morteza and Song, Jiaming and Kautz, Jan and Vahdat, Arash. ICLR 2024.
>
> [2] Pseudoinverse-guided diffusion models for inverse problems. Song, Jiaming and Vahdat, Arash and Mardani, Morteza and Kautz, Jan. ICLR 2023.
>
> [3] D-Flow: Differentiating through Flows for Controlled Generation. Heli Ben-Hamu, Omri Puny, Itai Gat, Brian Karrer, Uriel Singer, Yaron Lipman. ICML 2024 (concurrent work)

---

> > ### Comment · Reviewer_sT2G · 2024-08-11
> > **Response to the Rebuttal**
> >
> > Thank the authors for the rebuttal.
> >
> > I strongly recommend the authors to characterize more about the non-asymptotic case of c(N), given that the implementation is indeed not exactly the same as the assumption made in asymptotic case. The authors should explicitly clarify this in the paper, especially in the Theorem part. I have no question about the comparison with the traditional methods and the experiment settings. I appreciate the authors for their hard work on the additional experiments and clarification. After careful consideration, I keep the original rating, but increase the contribution rating and my confidence.

---

> ### Author Response · Authors · 2024-08-12
>
> We greatly appreciate your confidence in our work and your thoughtful suggestions regarding our theoretical contributions.   We promise to incorporate the necessary adjustments in the revision.  **Before the discussion period ends,  we would like to provide a last-minute  highlight of  the strengths of our approach, particularly in comparison to the concurrent flow-based method, D-Flow [1], as recognized by other reviewers.**  As shown in the Table 3 in the appendix of our paper, our method requires 1.6 min for each image, OT-ODE requires 2.46 min, while the concurrent work D-Flow [1], which formulates the MAP into a constrained optimization problem (Eq. 9), requires 5-10 mins for each image as documented in their Sec. 3.4. This is because each of its optimization steps requires a backpropogation through an ODE solver, as they calculate the log-likelihood fully. Our work is significantly faster due to our principled local MAP approximation. We hope this runtime comparison further highlights the novelty of our method.
>
> [1] D-Flow: Differentiating through Flows for Controlled Generation. Heli Ben-Hamu, Omri Puny, Itai Gat, Brian Karrer, Uriel Singer, Yaron Lipman. ICML 2024 (concurrent work)

---

### Official Review · Reviewer_Xm5z · 2024-07-13

**Soundness:** 3
**Presentation:** 3
**Contribution:** 3
**Rating:** 7
**Confidence:** 1

**Summary:**

This paper addresses the challenge of solving linear inverse problems in high-resolution image synthesis using generative models based on flow matching. While these models are appealing due to their ability to compute image likelihoods directly from a learned flow, they suffer from slow log-likelihood computations that involve backpropagation through an ODE solver. This computational bottleneck can be particularly problematic for high-dimensional problems.

To overcome this issue, the authors propose a novel iterative algorithm designed to approximate the maximum-a-posteriori (MAP) estimator efficiently. The key insight is to decompose the MAP objective into multiple "local MAP" objectives, leveraging Tweedie’s formula to perform gradient steps for sequential optimization. This approach allows for a more efficient approximation of the MAP estimator.

The paper validates the proposed algorithm across various linear inverse problems, such as super-resolution, deblurring, inpainting, and compressed sensing. The results show that the new method outperforms existing techniques based on flow matching, making it a promising tool for high-resolution image synthesis and related tasks.

The authors conclude that their approach successfully addresses the computational challenges of flow matching models, offering a robust solution for incorporating flow priors in solving linear inverse problems. They also mention discussing limitations and future work in the appendix.

**Strengths:**

- originality: the authors present an original idea to utilize the image probabilities obtained from the learned flows in Flow Matching as priors for MAP estimation in inverse problems
- quality: The authors provide a good account of Flow Matching and deep understanding of the inverse problems. Delivery of the argument is elaborated and supported with theoretical proofs as well as intuitive toy data examples and more complex tasks. Ablation studies to support the argument are provided.
- clarity: fairly clear, improvements on the flow of argument would be desired.
- significance: applications to real-world imaging problems such as medical imaging are very important, and the authors provide promising results in section 4.2, including Table 1 and Figure 4 and Appendix G.

**Weaknesses:**

The the paper could benefit from more clear presentation, tying the proofs and experimental results to the claims introduced at the beginning a bit tighter.

**Questions:**

NA

---

> ### Author Rebuttal · Authors · 2024-08-02
>
> We appreciate your recognition of our original contributions and will work on enhancing the clarity and coherence of our presentation. In the attached pdf above, more experiments have been added to show our method's capability to handle more challenging conditions ($\nu=1/10$, Tab. 1), highly noisy settings (Tab. 2), strong performance over classical signal recovery algorithms (Tab. 1) brought by utilizing generative models as priors, and its competitiveness with representative diffusion-based methods (Tab. 2).

---

### Official Review · Reviewer_hhXk · 2024-07-14

**Soundness:** 3
**Presentation:** 3
**Contribution:** 2
**Rating:** 5
**Confidence:** 4

**Summary:**

The paper proposes efficiently recovering MAP estimates by utilizing flow-matching priors. The main proposition in the presented method is to break down the MAP objective into a sum of N local MAP objectives, facilitating a computationally feasible approach that runs in reasonable time. The results are compared to several proposed baselines in terms of distortion (PSNR/SSIM) showing consistent improvements across two datasets and multiple tasks.

**Strengths:**

*    The paper proposes a theoretically motivated approximation of MAP estimates utilizing flow-matching priors.
*    The paper is well written and overall well structured.
*    The experiments entail multiple tasks and datasets, consistently showing improved performance in distortion.

**Weaknesses:**

*    The paper argues that flow-matching priors are useful due to the ability to calculate reconstruction log-likelihood, yet this information is never used and does not appear in the experiments section
*    The authors focus only on distortion (and not on perceptual quality), arguing for a single (blurry) MAP estimate. Nonetheless, by this point it is well understood within the image restoration community that summarizing posteriors to a single prediction is inevitably throwing away uncertainty information which is vital for proper down-the-line decision-making in safety-critical applications. Therefore, posterior samplers (aka stochastic inverse problem solvers) are a much better solution in underdetermined inverse problems with a multitude of admissible solutions.
*    Nowadays posterior samplers utilizing distilled diffusion models can produce samples with a single neural function evaluation. Compared to such methods, it is unclear what might be the advantage of the presented technique besides maybe accompanying each reconstruction with its likelihood, which is again missing in the presented results.

**Questions:**

*    What information is there to be gained from the likelihood computation if all the user is going to be presented with is a single MAP estimate?
*    In terms of runtime, how does your method fair with other types of MAP estimates employing different generative models as priors?
*    What changes will it take to apply your method to non-linear inverse problems?

**Limitations:**

The authors were upfront about the limitations of their method and stated these in Appendix A.

---

> ### Author Rebuttal · Authors · 2024-08-06
>
> We thank the reviewer for their detailed feedback and comments. We would like to address your concerns regarding our paper's motivation and contribution to the inverse problems community. We will respond to each concern below:
>
> **The use of log-likelihood and flow prior**
>
> **First**, we would like to note that our algorithm does use the likelihood under a flow prior. In particular, our algorithm optimizes a sequence of objectives that locally approximate the log-likelihood for each timestep $t$. In the lines 7 and 12 of Algo. 1 of the paper, the terms $\frac{1}{2}||x_t||$, $\log p(x_t)$, and the trace approximation are derived from the flow prior's log likelihood. We can obtain the gradient of $\log p(x_t)$ using Eq (11). We experimentally demonstrate that these prior terms are important in obtaining quality reconstructions in Fig. 2 by comparing our results to the results of ours without the local prior terms. Hence the flow prior and log-likelihood is heavily used in our algorithm, and are crucial to help us obtain good performance.
>
> **Second**, with flow-based models, we are able to calculate $\log p(x)$ by the instantaneous change-of-variable formula (Eq. 8 in the paper). We choose not to present the log-likelihoods and mainly present reconstruction metrics, such as PSNR and SSIM, to properly compare our method with flow-based baselines such as D-Flow [1] and OT-ODE [2], and  diffusion-based methods like RED-Diff [3] and $\Pi$GDM [4].  We fully agree that reporting log-likelihoods could help users gain more information of the reconstructed images and we promise to do so in the revision.
>
> **On MAP Estimation and Uncertainty Quantification** Thank you for raising this issue. We acknowledge that posterior sampling methods have the advantage that they can provide multiple reconstructions to visualize uncertainty. However, most methods that we compare to, such as our baselines of [1] [2] [3] and [4], only provide one estimate for each single image and do not focus on uncertainty quantification. We would like to highlight, however, that it is possible to equip MAP with uncertainty quantification, which we hope to explore in future work. For example, given an MAP solution $\hat x_{MAP}$, one can compute a Laplace approximation $p(x_{MAP}|y) \sim N(\hat x_{MAP}, H^{-1})$ where $H$ is the Hessian matrix of $\log p(x|y)$ at $\hat x_{MAP}$. This method has its pros and cons relative to posterior sampling methods. Here, computing this Hessian can be challenging, especially for flow-based log priors, but once calculated, sampling is straightforward. Similarly, posterior sampling methods require many iterations to generate many quality samples, which can be time-consuming. We believe investigating the benefits of this approach constitutes an interesting direction for future work.
>
> **On Distilled Models** While distilled models offer a promising way to speed up inversion with diffusion models, to our knowledge there have been few works that have successfully demonstrated their strong performance. Moreover, one-step distilled diffusion models behave similarly to push-forward generators, such as GANs. Even with distilled diffusion models, both posterior sampling and MAP estimation require many optimization steps to achieve satisfactory reconstructions, as shown in classical papers using GAN priors. See, for example, Sec 6.1.1 in CSGM [5], where they optimize the latent space of the GAN for 1000 steps to reconstruct an MNIST digit.
>
> **Runtime Comparison** As shown in the Table 3 in the appendix of our paper, our method requires 1.6 min for each image, OT-ODE [2] requires  2.46 min, while the concurrent work D-Flow [1], which formulates the MAP into a constrained optimization problem (Eq. 9), requires 5-10 mins for each image as documented in their Sec. 3.4. This is because each of its optimization steps requires a backpropogation through an ODE solver, as they calculate the log-likelihood fully. Our work is significantly faster due to our principled local MAP approximation.
>
>  Finally, we want to highlight our theoretical contribution to inverse problems with flow-based models as priors, where backpropagation through an ODE solver (usually requiring 100 or more NFEs) is avoided, and asymptotic convergence is proven, as demonstrated in Theorem 1. Practically, our ICTM algorithm consistently achieves strong performance in distortion across multiple tasks and datasets, from natural images to medical application, as evidenced by Tabs. 1 and 2 in the attached pdf above. We kindly request a re-evaluation of the rating if some of your concerns have been resolved.
>
> [1] D-Flow: Differentiating through Flows for Controlled Generation. Heli Ben-Hamu, Omri Puny, Itai Gat, Brian Karrer, Uriel Singer, Yaron Lipman. ICML 2024. (concurrent work)
>
> [2] Training-free linear image inverses via flows. Ashwini Pokle, Matthew J. Muckley, Ricky T. Q. Chen, Brian Karrer. TMLR 2024.
>
> [3] A Variational Perspective on Solving Inverse Problems with Diffusion Models. Mardani, Morteza and Song, Jiaming and Kautz, Jan and Vahdat, Arash. ICLR 2024.
>
> [4] Pseudoinverse-guided diffusion models for inverse problems. Song, Jiaming and Vahdat, Arash and Mardani, Morteza and Kautz, Jan. ICLR 2023.
>
> [5] Compressed Sensing using Generative Models. Ashish Bora, Ajil Jalal, Eric Price, and Alexandros G. Dimakis. ICML 2017.

---

> ### Author Response · Authors · 2024-08-12
>
> Dear reviewer,
>
> Thanks a lot for your time and effort. As the discussion period approaches its deadline, we would greatly appreciate your feedback on whether our rebuttal has adequately addressed your concerns. Please reach out if you have additional questions or comments which we are happy to address. Thank you again.
>
> Best,
> Authors

---

> > ### Comment · Reviewer_hhXk · 2024-08-12
> >
> > Thanks for the clear and thorough rebuttal. I'm hesitant to change my score mainly because I do not agree with you on two points:
> > * First, the likelihood information, if the user is to be presented with a single reconstruction, is practically useless as ultimately what matters is the likelihood ratio between different possibilities.
> > * Second, the results are extra blurry. As evident in the FID numbers from the rebuttal PDF, this method is hardly competitive in perceptual quality.
> >
> > I do recognize the advantage of this method over the concurrent work of D-Flow. Nonetheless, in its current form, I think it has limited applicability and therefore I maintain my score.

---

> ### Author Response · Authors · 2024-08-13
>
> We greatly appreciate your time and recognition of our paper's strengths compared to the concurrent work, D-Flow. We would like to provide a last-minute clarification on the two points you mentioned, which we hope will address some of your concerns:
>
> - **Regarding likelihood information:** We appreciate that giving the user likelihood estimates in a more systematic way could be useful, but we want to highlight that the main focus of our work is using a flow-based prior's likelihood to help obtain high-quality reconstruction estimates for inverse problems in a computationally efficient way. As compared to baselines such as D-Flow, we can do this in a computationally efficient manner based on our theoretically-motivated MAP approximation. Moreover, focusing on reconstruction enables a fair comparison with flow-based baselines such as D-Flow and OT-ODE, as well as diffusion-based methods like RED-Diff and $\Pi$GDM. Of note is that likelihood information has also not been provided in the experiments of the four baselines.
>
> - **Perceptual quality:** We would like to highlight that our method demonstrates strong competitiveness in perceptual quality, as evidenced by the fact that our method outperforms $\Pi$GDM in terms of FID in super-resolution and inpainting (random), surpasses OT-ODE in both FID and LPIPS in inpainting (random) and inpainting (box), and exceeds RED-Diff in both FID and LPIPS across all tasks except Gaussian Deblurring. As shown in the Figure 1 in the rebuttal pdf, our method generates clear images even in the highly noisy setting and is strong in terms of PSNR and SSIM. Overall, our method achieves consistently improved performance in reconstruction while maintaining competitive perceptual quality compared to the baselines.

---

> > ### Comment · Reviewer_hhXk · 2024-08-13
> >
> > Thank you for these additional clarifications. After reading the reviews of others, re-reading the paper, and looking again into the rebuttal PDF, I would like to raise my score to 5 to enhance your chances of getting accepted. Best of luck!

---

> > > ### Author Response · Authors · 2024-08-13
> > >
> > > We greatly appreciate the time and effort you've dedicated to our work. Thank you for your support and thought-provoking suggestions!

---

### Author Rebuttal · Authors · 2024-08-05

Dear Reviewers,

We first thank you for your valuable feedback and appreciate the recognition of our paper's strengths, such as

- concise presentation from concepts to methods and clear motivation (sT2G),
- a solid theoretical foundation supporting the method (qjx2),
- consistently improved performance in distortion across multiple tasks and datasets (hhXk).

In the attached pdf, we provide additional experimental results that we hope address your concerns (implementation details are documented at the end):


**[Highlight: more diffusion-based baselines in Table 2]**
In addition to the flow-based baselines, we have included two representative diffusion-based baselines: 1) RED-Diff [1], a variational Bayes-based method; and 2) $\Pi$GDM [2], an advanced MCMC-based method. Our method demonstrates competitive performance with both diffusion-based baselines and other approaches in terms of recovery fidelity. Additionally, one reviewer mentioned a concurrent work, D-Flow [3], recently accepted to ICML 2024, which has not yet released its official code. Notably, D-Flow also includes [1] and [2] as diffusion-based baselines.



**Table 1(two classical recovery algorithms and a more challenging setting $\nu=1/10$ included)**: Results of compressed sensing with varying compression rate $\nu$ on the HCP T2w dataset. We have added results for two classical recovery algorithms, Wavelet and TV priors, as well as a more challenging setting  $\nu=1/10$, where only 10\% of the output signal is observed. Our method outperforms the classical recovery algorithms and other baselines in all settings, demonstrating our method's capability to handle challenging scenarios and the advantages of utilizing modern generative models as priors.




**Table 2(two new baselines and a highly noisy setting $\sigma_y=0.2$ included)**: Quantitative comparison results of PSNR and SSIM on the CelebA-HQ dataset (best values highlighted in blue and second-best underlined). We have included two representative diffusion-based baselines: 1)RED-Diff [1], a variational Bayes-based method; 2)$\Pi$GDM [2], an advanced MCMC-based method. Our method is competitive with diffusion-based baselines and other approaches in terms of recovery fidelity. Additionally, we included an extremely noisy setting with
 $\sigma_y=0.2$.  The last row of the table shows our method's capability to handle noisy settings across different tasks.


**Table 3(FID and LPIPS scores included)**: Quantitative comparison results of FID and LPIPS on the CelebA-HQ dataset (best values highlighted in blue and second-best underlined). This table shares the same setting as Table 2. Our method is also competitive with other baselines in terms of perceptual quality as evidenced by LPIPS.

Please see our reviewer-specific feedback for more information.  We kindly request a re-evaluation of the rating if some of your concerns have been resolved.

[1] A Variational Perspective on Solving Inverse Problems with Diffusion Models. Mardani, Morteza and Song, Jiaming and Kautz, Jan and Vahdat, Arash. ICLR 2024.

[2] Pseudoinverse-guided diffusion models for inverse problems. Song, Jiaming and Vahdat, Arash and Mardani, Morteza and Kautz, Jan. ICLR 2023.

[3] D-Flow: Differentiating through Flows for Controlled Generation. Heli Ben-Hamu, Omri Puny, Itai Gat, Brian Karrer, Uriel Singer, Yaron Lipman. ICML 2024 (concurrent work)

---
[Implementation Details]

Table 1: We use the pytorch package *DeepInverse*$^1$ to implement Wavelet and TV priors as shown in Tab.1 of the pdf. For both priors, we use the default Proximal Gradient Descent  algorithm and perform a grid search for regularization weight $\lambda$ in the set {$10^0, 10^{-1}, 10^{-2}, 10^{-3}, 10^{-4}$} and gradient stepsize $\eta$ in {$10^0, 10^{-1}, 10^{-2}, 10^{-3}, 10^{-4}$}. The maximum number of iterations is 3k, 5k, and 10k for compression rate $\nu = 1/2, 1/4,$ and $1/10$, respectively. The stopping criterion is the residual norm $\frac{||x_{t-1}-x_t||}{||x_{t-1}||} \le 1\times 10^{-5}$ and the initialization of the algorithm is the backprojected reconstruction, i.e., the pseudoinverse of $\mathcal{A}$ applied to the measurement $y$.

- For the TV prior, the objective we aim to minimize is $\min_x \frac{1}{2}||\mathcal{A}x - y||_2^2 + \lambda \||x\||{\rm tv}$.  We find that the optimal combination of hyperparameters is $\lambda = 0.01, \eta = 0.1$ for all values of $\nu$.

- For the Wavelet prior, the objective we want to minimize is $\min_x \frac{1}{2}||\mathcal{A}x - y||_2^2 + \lambda ||\Psi x||_1$. We use the default level of the wavelet transform and select the “db8” Wavelet. The optimal combination of hyperparameters is
$\lambda=0.1, \eta=0.1$ for all values of $\nu$.

Table 2: We use the official repository$^2$ from Nvidia to reproduce the results of RED-Diff and $\Pi$GDM with the pretrained CelebAHQ checkpoint using the architecture of the guided diffusion repo from OpenAI$^3$.

- For RED-Diff, the optimization objective is $\min_\mu  ||y - \mathcal A (\mu)||^2 + \lambda  (sg(\epsilon_\theta(x_t,t) - \epsilon))^T \mu$. Following the implementation of the original paper, we use Adam optimizer with 1,000 steps for all tasks. We choose learning rate $lr=0.25, \lambda=0.25$ for super-resolution, inpainting(random) and inpainting(box) and $lr=0.5, \lambda=0.25$ for deblurring as recommended by the paper.

- For $\Pi$GDM, we follow the original paper and use 100 diffusion steps. Specifically, we use $\eta = 1.0$ which corresponds to the VE-SDE. Adaptive weights $r_t^2 = \frac{\sigma_{1-t}^2}{1+\sigma_t^2}$ are used if there is an improvement on metrics.


1. https://deepinv.github.io/deepinv/
2. https://github.com/NVlabs/RED-diff
3. https://github.com/openai/guided-diffusion

---

### Author Response · Authors · 2024-08-10
**Kind reminder**

Dear all reviewers,

Thank you for all the insightful and constructive comments and feedback, which greatly helped us to improve the paper quality.

This is a kind reminder that the discussion period is closing in a few days. Feel free to check our response to your questions and tell us if you have any further concerns.

Best,
Authors

---

### Author Response · Authors · 2024-08-14
**A summary of the discussions**

Dear AC,

Thank you for your time and effort in evaluating our work. We would like to provide a brief summary of the discussions between the reviewers and us to facilitate a better evaluation. Overall, our responses to each reviewer's questions received positive feedback.

- **Reviewer hhXk:** After acknowledging our method's strength over the recent concurrent work, D-Flow, the reviewer expressed major concerns regarding likelihood information and perceptual quality in their second-to-last comment. We emphasized that our method focuses on computationally efficient, high-quality reconstructions with highly competitive perceptual performance compared to other baselines, as evidenced by our rebuttal PDF. Following our response, the reviewer stated, *raised my score to enhance your chances of getting accepted.*

- **Reviewer sT2G:** The reviewer’s main concerns were about the comparison with traditional methods, more challenging experiment settings, and the need for clearer specifications in our theorem statement. After our response, the reviewer had no further questions on the experiments required and stated, *increase the contribution rating and my confidence.*

- **Reviewer qjx2:** The reviewer acknowledged that we *have addressed my concerns by providing empirical evidence for the requested cases*, such as severely ill-posed inverse problems in highly noisy settings, with more baselines compared and perceptual metrics reported. Consequently, the reviewer chose to *raise the score to 5*.

We hope this summary helps you better evaluate the discussion process.


Best,
Authors

---

### Decision · Program_Chairs · 2024-09-25

**Decision:**

Accept (poster)

**Comment:**

The paper has received all positive reviews, with one strongly in favour but with low confidence and the rest borderline accepts. After the discussion phase, reviewers raised their score. Reviewers appreciate the theoretical contribution of integrating flow matching with into MAP for solving inverse problems is a well motivated and interesting approach, the good writeup and improved experimental results. On the other hand, some significant weaknesses also persist, including lack of some important metrics (perceptual quality), limited usefulness of likelihood information, limited number of baselines and experimental datasets and asymptotic error bound usefulness
In light of the positive rebuttal phase that successfully addressed some of the above-mentioned issues and the positive scores and the overall solid contribution, the paper is accepted.